# Disconnects between Dataset Representativeness and Group Algorithmic Fairness

## Abstract

There have been numerous demonstrations that the prediction performance of machine learning algorithms differs among groups of people. This causes significant concerns about long-term social impact, including the perpetuation of disadvantages for certain populations. A common explanation is that disparity in performance (i.e., group unfairness) results from differences in group representation in datasets. Recent research has started to explore this explanation and proposed methods to address group unfairness by modulating group representation. We establish that, contrary to conventional wisdom, there exists a fundamental tradeoff between representativeness and group fairness. First, we theoretically describe this tradeoff in a simple univariate setting and confirm our theoretic results empirically across several commonly used datasets. To analyze whether these observations hold in more realistic settings, we then model the process of constructing representative datasets from multiple data sources using a multi-armed bandit framework and a novel Bayesian approach. We find that realistic sampling techniques further nuance the relationship between dataset representativeness and fairness. Notably, we show how the theoretically-sound solution of oversampling groups with lower performance may not hold for realistic multi-site data collection. Finally, we postulate that a key driver of unfairness is the extent to which labels are more challenging to predict for some groups than others. To validate this hypothesis, we show that greater model capacity can lead to improved group fairness independently of representation. In summary, we demonstrate how representativeness and group fairness may be at odds, how theoretically justified approaches to improve fairness may not hold true under realistic conditions, and propose a representation-independent method to improve algorithmic fairness.

## 1 Introduction

As society increasingly relies on artificial intelligence (AI) models trained on aggregated datasets, ethical AI guidelines have stressed the importance of responsible data analysis (Ng et al., 2022). One facet of ethical AI development is the analysis of algorithmic fairness, commonly defined as (near)-equal treatment of different groups by an AI model (Mitchell et al., 2021). Numerous definitions have emerged to formalize group fairness, largely differing in the way that they compare performance among differing groups (Hardt et al., 2016; Kleinberg et al., 2017). Across one or more of these definitions, there is extensive literature documenting failures of group fairness (i.e., *unfairness*) of common learning algorithms (Buolamwini & Gebru, 2018; Mehrabi et al., 2021).

As it is common for relative performance disadvantage to manifest for groups who are underrepresented in datasets (Abernethy et al., 2020; Bentley et al., 2020; Feldman et al., 2015; Kearns et al., 2018; Shekhar et al., 2021; Sirugo et al., 2019), historic underrepresentation has come to be commonly seen as a key explanation for lack of fairness (Huppenkothen et al., 2020; Idrissi et al., 2022; Li et al., 2022; Mapes et al., 2020; Nargesian et al., 2021; Wang & Russakovsky, 2023; National Academies of Sciences et al., 2022). However, some recent evidence suggests that the relationship between representativeness and group fairness is more subtle. Following much literature in this space, representativeness is measured as the distance between the distribution of some attributes in a reference population and the population of the dataset (He et al.,

2016). Both Chen et al. (2018) and Li et al. (2022) have investigated the relationship between group fairness and several data-centric factors, including representativeness. Increasing representation of underrepresented subgroups sometimes, but not always, leads to greater fairness and these analyses remain somewhat limited: Li et al. (2022) considers only one dataset, while Chen et al. (2018) does not investigate the full spectrum of group representation.

A more fundamental limitation of existing works is that they treat representativeness as a directly tunable parameter, typically by adding or randomly removing examples within a particular group (Chen et al., 2018; Li et al., 2022). These methodologies assume that a dataset collector can directly control the relative proportions of groups without fundamentally affecting group-conditional distributions. This assumption holds true for random data removal but discarding examples may be undesirable when their acquisition has significant costs, as is often the case with recruitment. More importantly, this assumption may not hold true when adding additional data. Single data sources are rarely representative of the true reference population (Borza et al., 2022; Mapes et al., 2020), so representative datasets are often constructed from multiple data sources. For example, the *All of Us* Research Program collects data from numerous recruitment centers across the United States (Denny et al., 2019). We hypothesize that datasets constructed from multiple sites will demonstrate a different representativeness-fairness relationship than datasets that are subsampled from a larger data domain. Thus, two datasets from the same domain with equal representativeness may still vary substantially in their feature distributions, especially for attributes that representativeness is not measured over. This variation may yield distinct representativeness-fairness relationships.

In this paper, we explore the tension between representativeness and fairness through theory, idealized experimental scenarios, and realistic experimental scenarios. We then draw upon our theoretic results to propose an alternative method that improves group fairness independent of subgroup representation, namely increasing model capacity. Our paper is organized as follows:

- First, Section 2 formalizes definitions of representativeness and fairness for this work.

- In Section 3, we present theoretic results establishing a tension between representation and fairness with a case study in single variable classifiers.

- Section 4 outlines our experimental methodology and summarizes our strategies for realistic site-based sampling.

- Section 5 provides empiric results validating the tension described in Section 3.

- Extending these results, Section 6 demonstrates how the choice of sampling methodology complicates the representation-fairness relationship.

- Section 7 conversely shows how fair sampling algorithms yield unrepresentative datasets.

- Lastly, Section 8 studies the ability of model complexity to mitigate unfairness independent of group representation.

## 1.1 Related Work

Algorithmic fairness literature has considered a broad array of formal notions of fairness. A classic definition is individual fairness, where similar individuals should receive similar outcomes (e.g., predictions) (Dwork et al., 2012). A common alternative notion is one of *group fairness*, where the goal is to achieve equity among groups in terms of common efficacy metrics such as accuracy, false positive rate, false negative rate, and calibration (Diana et al., 2021; Mitchell et al., 2021); in general it is impossible to achieve all of these simultaneously (Kleinberg et al., 2017). Moreover, a number of approaches aim to ensure group fairness in one or more of these metrics by either training fair models, or a post-hoc adjustment to predictive models (Hardt et al., 2016; Barocas et al., 2023). These approaches typically take a dataset as given.

In addition to the issue of fairness, considerable work has been devoted to the problem of constructing representative datasets, commonly captured in terms of statistical distance of the attribute distribution in a dataset from a target distribution (He et al., 2016; Celis et al., 2020; Qi et al., 2021; Asudeh et al., 2019; Jin

et al., 2020). Given such measures, several approaches have been proposed for constructing representative datasets, typically in ways that are task-specific (Huppenkothen et al., 2020; Flanigan et al., 2021; Borza et al., 2022; Nargesian et al., 2021).

Some recent work has also used notions of group fairness to guide active data collection (Abernethy et al., 2020; Shekhar et al., 2021; Niss et al., 2022). However, these approaches require prior knowledge about the downstream tasks to be performed on collected data. General purpose datasets like the *All of Us* Research Program support a multitude of model types and predictive tasks, most of which are unknown at the time of data collection (Denny et al., 2019). Post-hoc group rebalancing may improve downstream algorithmic fairness to the same extent as optimization-based methods (Idrissi et al., 2022; Zhang et al., 2022), but post-hoc corrections may severely impact predictive accuracy (Woodworth et al., 2017).

The relationship between representation and fairness is less explored than its two constituent concepts. As noted above, classifier performance tends to be poorer for underrepresented groups than overrepresented groups. Increasing representation of these groups in training or finetuning data can sometimes, but not always, reduce fairness disparity (Wang & Russakovsky, 2023; Li et al., 2022). Chen et al. (2018) provide a framework to understand this apparent inconsistency by decomposing algorithmic unfairness (i.e., discrimination) into bias, variance, and noise terms. Unfairness driven by bias stems from a misspecified model hypothesis class that better suits one protected group, unfairness driven by variance accounts for dataset-dependent prediction differences, and unfairness driven by noise refers to unobserved variables affecting the dataset labels. Of these contributors to unfairness, only variance depends on the composition (and possible representativeness) of the dataset. Chen et al. (2018) show how sampling more training data, especially of harder-to-predict groups, may improve fairness. However, the empirical analyses of Chen et al. (2018) and Li et al. (2022) do not comprehensively analyze the effects of group representation on fairness. Celis et al. (2020) use an underlying dataset to generate maximum entropy distributions from which to sample pseudo-data that satisfy both outcome fairness and a desired level of representativeness. While their methods are relatively efficient, pseudo-data may not always be practical to generate or desired for downstream tasks.

## 2 Preliminaries

Let $\mathcal{X} \subset \mathbb{R}^d$ be a domain of features, $\mathcal{A} \subset \mathbb{R}^\ell$ be a domain of sensitive attributes (e.g., race or gender), and $\mathcal{Y} \equiv \{0, 1\}$ be binary labels. We investigate the problem of building a dataset $(X, A, Y) \subset \mathcal{X} \times \mathcal{A} \times \mathcal{Y}$, specifically focusing on the *representativeness* and *fairness* of that dataset.

**Definition 1. (Representativeness):** *The representativeness of a dataset $(X, A, Y)$ with respect to a target demographic vector $\boldsymbol{v} \in \mathbb{R}^\ell$ and distance metric $\mathcal{M}$ is inversely proportional to $\mathcal{M}\left(\boldsymbol{v}, \frac{1}{|A|} \sum_{\boldsymbol{a} \in A} \boldsymbol{a}\right)$, where $\frac{1}{|A|} \sum_{\boldsymbol{a} \in A} \boldsymbol{a}$ is the mean vector of the demographics in the dataset.*

Suppose there are two binary features of interest: gender (Male or Female) and age (Young or Old). A target vector of $\boldsymbol{v} = \langle 0.3, 0.7 \rangle$ implies that a representative dataset ($\mathcal{M} = 0$) is 30% Male and 70% Young. If $\mathcal{M}$ is the $\ell_1$-norm, then a dataset which is 25% Male and 60% Young would be 0.15-distant with respect to $\boldsymbol{v}$.

**Definition 2. (Fairness):** *For a given dataset $(X, A, Y)$ and classifier $f : \mathcal{X} \to \mathcal{Y}$, and performance measure $U$, the* unfairness *of $f$ on group $\mathbf{a} \in A$ is*

$$\left| U\big(f, X, Y, A)\big) - U_{\mathbf{a}}\big(f, X, Y, A\big) \right|$$

*Where $U_{\mathbf{a}}$ is the measure $U$ applied to only examples with sensitive feature $\mathbf{a}$.*

## 3 Univariate Representation and Fairness

We begin our investigation into the relationship between representatives and fairness with an illustrative example in the case of univariate features.[1] Within this example, we demonstrate that there is a fundamental tension between representativeness and fairness; we later verify this relationship empirically.

---

[1]Univariate and multivariate classification are equivalent in the sense that $x$ can correspond to the output of a score function applied to the multidimensional feature $\mathbf{x}$, i.e. $x = h(\mathbf{x})$.

Let there be a binary sensitive feature $\mathcal{A} \equiv \{0, 1\}$. We begin by demonstrating the existence of a trade-off between fairness and representativeness. This trade-off stems from the relative difference in *difficulty* of learning the conditional distribution $\mathbb{P}(y = 1|x, A = a)$; as the difference in difficulty increases between groups, the tradeoff between representativeness and fairness becomes more severe. To capture the difficulty of learning this distribution, let the relationship between $x$ and $y$ be defined as $y = \mathbb{I}[x + \varepsilon_g \geq \theta_g]$ where $\varepsilon_g \sim \mathcal{N}(\mu_g, \sigma_g)$ gives the noise of the label $y$. Let $D_g$ be the distribution over features and labels for group $g$, and let $\mathcal{F}$ be a group-aware classifier acting on features $x$.

**Theorem 1.** *Suppose there are $n_0$ and $n_1$ samples collected from groups $g = 0$ and $g = 1$ respectively. Let $\mathcal{F}$ be the optimal classifiers learned on these samples (in terms of expected accuracy). Let $\delta = error(\mathcal{F}, D_0) - error(\mathcal{F}, D_1)$, i.e., the difference in accuracy between groups. Then*

$$\mathbb{E}[\delta] = \sqrt{2/\pi}\left(\sigma_0\sqrt{1/n_0} - \sigma_1\sqrt{1/n_1}\right)$$

*Proof.* Let $(x, y)$ be one datapoint, i.e., a feature and label respectively. Suppose that for a given $x$ the label $Y$ is induced via $y = \mathbb{I}[x + \varepsilon_g \geq \theta_g]$ where $\varepsilon_g \sim \mathcal{N}(\mu_g, \sigma_g)$. Let $(\mathbf{X}, \mathbf{Y})$ be a dataset of $n_0$ such examples from group $g_0$ and $n_1$ such examples from group $g_1$. Let $\mathcal{F}$ be the classifier with the highest accuracy on $(\mathbf{X}, \mathbf{Y})$.

Then the expected unfairness of $\mathcal{F}$ with respect to each group's true distribution over features and labels $D_g$, can be written as

$$\delta = \text{error}(\mathcal{F}, D_0) - \text{error}(\mathcal{F}, D_1)$$

In the case that $\mathcal{F}$ is a threshold classifier acting on both groups, the classifier with the highest accuracy on data $(\mathbf{X}, \mathbf{Y})$ will have the propriety that

$$\mathcal{F}(x|g) = 1 \ \text{ if } \ x \geq 1/n_g \sum_{x_j \in \mathbf{X}|g} x, \quad \text{ and } 0 \ \text{ otherwise}$$

where $1/n_g \sum_{x_j \in \mathbf{X}|g} x$ is the mean value of all features in $\mathbf{X}$ which correspond to group $g$. Thus, each error term $\text{error}(\mathcal{F}|g)$ is proportional to the empirical mean $1/n_g \sum_{x_j \in \mathbf{X}|g} x$ and the true feature mean $\theta_g$. By the Mean Absolute Difference for normal distributions, this value is

$$\sigma_g \sqrt{2/\pi n_g}$$

for each group. Thus the expected difference in error rates is $\mathbb{E}[\delta] = \sqrt{2/\pi}\left(\sigma_0\sqrt{1/n_0} - \sigma_1\sqrt{1/n_1}\right)$ $\qquad \square$

The key takeaway from Theorem 1 is that it allows us to quantify expected unfairness $\mathbb{E}[\delta]$ in terms of both the number of samples collected from each group $n_0, n_1$ and the relative noisiness of each groups' labels $\sigma_0, \sigma_1$. The expression of expected unfairness immediately yields the following:

**Theorem 2.** *Suppose the optimal classifier trained on $n_0$ samples from group $0$ and $n_1$ samples of group $1$ has an unfairness of at most $\delta$, then it must be the case that*

$$n_1\left(\frac{\sigma_0}{\delta\sqrt{\pi/2n_1} + \sigma_1}\right)^2 \leq n_0 \ and \ n_0\left(\frac{\sigma_1}{\delta\sqrt{\pi/2n_0} + \sigma_0}\right)^2 \leq n_1$$

*Proof.* By Theorem 1, having an unfairness of size at most $\delta$ requires

$$-\delta \leq \sqrt{2/\pi}\left(\sigma_0\sqrt{1/n_0} - \sigma_1\sqrt{1/n_1}\right) \leq \delta$$

By first examining the left-side inequality with respect to group $g_1$ we get

$$\sigma_1\sqrt{1/n_1} \leq \delta\sqrt{\pi/2} + \sigma_0\sqrt{1/n_0}$$

$$\Rightarrow 1/n_1 \leq \left(\frac{\delta\sqrt{\pi/2} + \sigma_0\sqrt{1/n_0}}{\sigma_1}\right)^2$$

$$\Rightarrow n1 \geq \left( \frac{\sigma_1}{\delta\sqrt{\pi/2n_0} + \sigma_0} \right)^2 n_0$$

A similar algebraic reduction when examining the right-side inequality with respect to group $g_0$ yields the other inequality. $\square$

This theorem indicates that, in order to limit the accuracy-disparity between groups to be no greater than $\delta$, the number of samples collected from each group $(n_0, n_1)$ cannot be too different.

**Theorem 3.** *In order to achieve an unfairness of* 0*, the sample ratio between the two groups must be* $n_0 = (\sigma_0^2/\sigma_1^2)n_1$.

*Proof.* This result follows directly from Theorem 1. $\square$

This theorem demonstrates that achieving an expected unfairness close to 0 may not be possible within a budget of $m$ total samples (i.e., $m = n_0 + n_1$). To see this, imagine a case in which $\sigma_0^2/\sigma_1^2 > m + 1$, i.e., group 0 has vastly higher noise than group 1. Then, the sampler will not be able to collect enough samples to ensure that $(\sigma_0^2/\sigma_1^2)n_1 < n_0$.

## 4 Experimental Methodology

In this section, we describe our general experimental methodology to investigate the relationship between fairness and representativeness. Data processing details and code are available in appendix B.

### 4.1 Domains

Our evaluation throughout considers six commonly-used datasets with natural sites and/or significant unfairness by sensitive features. To avoid overloading the term "dataset", we refer to these datasets henceforth as *underlying-datasets* or *domains*: 1) **Law School** (Wightman & Ramsey Jr, 1998), 2) **Lending Club** (Club, 2020), 3) **Intensive Care** (Pollard et al., 2018), 4) **Texas Salary** (Tribune, 2021), 5) **Adult Income** (Becker & Kohavi, 1996), and 6) **Community Crime** (Redmond, 2009). Each data domain contains sensitive features of interest (Tab. 1), which are binarized via one-hot encoding if not already binary. For consistency, we label the less prevalent group as $\boldsymbol{A} = 0$ $(G_0)$ and the more prevalent group $\boldsymbol{A} = 1$ $(G_1)$ across all sensitive features and domains. If a uniform target vector $v = \langle 0.5, \cdots, 0.5 \rangle$ is used, $G_0$ is underrepresented and $G_1$ is overrepresented.

| Data Domain | Sensitive Features | Target Feature | Location | Size (filtered) |
|---|---|---|---|---|
| Law School | Race, Gender, Age, Family Income | Pass Bar | School | 20,454 (16,950) |
| Lending Club | Housing Status, Occupation | Repay Loan | ZIP Code | 124,040 (115,368) |
| Intensive Care | Race, Gender, Age | ICU Recovery | Admission Type | 48,612 (48,259) |
| Texas Salary | Race, Gender | Earn $\geq$ \$75k | Office | 142,981 (121,210) |
| Adult Income | Race, Gender, Age | Income $\geq$ \$50k | — | 46,447 |
| Community Crime | Race Proportion | Low Crime Risk | — | 1,994 |

Table 1: Data Domains. All sensitive and target features are binarized. Filtering refers to the exclusion of sites with $< 2,000$ records (see section 6 for further details).

### 4.2 Dataset Construction

To study the relationship between fairness and representativeness in dataset construction we examine constructing a dataset of $n$ examples via three types of sampling: direct sampling, site-based sampling, and distributed site-based sampling.

**Direct sampling** is the least constrained, and least realistic, type of sampling. The data collector has direct access to the data domain and simply needs to select the $n$ examples that best approximate the target population. We implement direct sampling via stratified random sampling (SRS).

**Site-Based Sampling** models the more realistic scenario where data collectors lack direct access underlying-dataset and must construct their dataset by iteratively sampling from *sites* (Theodorou et al., 2024). Each site $s_i$ has a distribution of features and labels over a partition of the underlying data, but this distribution (as well as the distribution of the underlying-dataset) are unknown to the data collector until collection begins. Over the course of $T$ timesteps, the data collector iteratively selects a site and receives $k = n/T$ samples according to the distribution of that site. Sites are defined by the *Location* variable outlined in Table 1.

To analyze the effects of site-based sampling, we use several algorithms. We use two state-of-the-art sampling algorithms for convex objectives:[2] **UCB-LCB** [UCB] (Agrawal & Devanur, 2014) and **OL-Vec** [VEC] (Kesselheim & Singla, 2020). UCB-LCB is an adaptation of the classic UCB algorithm to handle convex objectives. OL-Vec solves site-based sampling as an online convex optimization problem. In addition to these two state-of-the-art algorithms, we also use two algorithms of our own creation: **PRBS** (appendix A.3), which is an adaptation of Thompson sampling to the case of convex objections, and **UCB-BY** (appendix A.3) which is an adaption of UCB-LCB that makes use uses of Bayesian priors. Additionally, we use $\varepsilon$-*Greedy* [$\varepsilon$GRD], and random sampling [RND] as a naïve baseline. To serve as gold standards, we also use full-information myopic optimal sampling algorithm **OPT**. For full details of algorithm implementation, see appendix A.

**Distributed Site-Based Sampling** adapts site-based sampling to situations where data collectors may simultaneously sample from multiple sites at each time step $t$ (Borza et al., 2024). For example, a data collector with budget $k = 6$ could collect 4 samples from site $s_1$ and 2 samples from site $s_5$. We use **D-PRBS** (appendix A.3),[3], a distributed modification of **PRBS** for distributed sampling. Additionally, we generate cohorts with **D-OPT**, a distributed version of OPT.

### 4.3   Measures

**Representativeness:** To measure representativeness of a dataset $(X, A, Y)$ with respect to a target demographic vector $\mathbf{v}$, we use $\mathcal{M}(\mathbf{v}, A) = \left\| \frac{1}{|A|} \sum_{\mathbf{a} \in A} \mathbf{a} - \mathbf{v} \right\|_2^2$.
**Fairness:** To measure the fairness of a dataset $(X, Y, A)$ and Gradient Boosted Classifier (GBC), we fit the classifier to $(X, Y)$ and evaluate fairness on a hold-out set. By analyzing group-wise differences in True Positive Rate (TPR), True Negative Rate (TNR), and Area Under the Curve (AUC), we average fairness across ten splits (90% train, 10% hold-out). By assessing TPR and TNR, we also assess their complements of False Negative Rate and False Positive Rate, respectively.

## 5   Effects of Data Composition on Fairness

In our first set of experiments, we assess the impact of group proportions on both classifier fairness and group-wise classifier performance under direct sampling (i.e., SRS). The size of each training dataset is set to the size of the smallest sensitive feature group, i.e., the largest dataset that may be constructed with 100% $G_0$. We present results for the Adult income dataset and defer other datasets (which are qualitatively similar) to appendix C.2. For each experiment, we down-sample the data domain using SRS, such that the resulting dataset has a specified fraction of members from $G_0$, defined in the Adult Income data domain either via race, gender, or age. As noted, results are reported as the average (and 95% CI) performance of 10 classifiers, corresponding to the 10 data domain splits.

We highlight salient empiric results studying the relationship between representation and fairness when stratified random sampling is used to select the training dataset. Full results on all relevant datasets may be

---

[2]Note that when $\mathcal{M}$ is convex, such as $\mathcal{M} = \|.\|_2^2$, so is the objective of representative sampling.
[3]To the best of our knowledge, there does not exist algorithms for distributed site-based sampling with convex objective; we thus rely only on our algorithm D-PRBS and the myopic optimal D-OPT in the distributed setting.

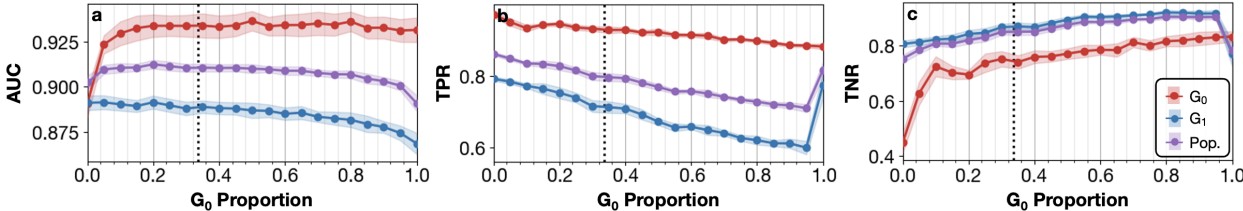

Figure 1: There is significant AUC (**a**), TPR (**b**), and TNR (**c**) unfairness by gender in the Adult Income data domain. Population (purple) and subgroup (red and blue) AUCs, TPRs, TNRs, and 95% CIs (shaded) are plotted for varying $G_0$ proportions. The dotted black line indicates the test set $G_0$ proportion.

found in appendix C. The Adult Income data domain shows significant AUC and TPR unfairness across all three tested sensitive features: gender (Fig. 1), as well as race and age (Appendix C.2). Notably, in Figure 1, increasing the proportion of $G_0$ in the training dataset results in worse TPR and better TNR for *both* groups. These opposing effects mean that modifying proportions of groups $G_0$ and $G_1$ has limited effect on subgroup AUC, except at the extremes (i.e., group proportions of 0 or 1). Fairness analyses of the Adult Income data domain using race as the sensitive feature shows similar TPR and TNR trends as a function of $G_0$ proportion, but using age as the sensitive feature reverses the trends (Appendix C.2). Thus, the choice of fairness metric and choice of sensitive feature can result in a qualitatively different relationship between fairness and representativeness.

## 6    Impact of Realistic Sampling Techniques

While stratified random sampling is useful to isolate the effect of group proportions on fairness, SRS is not reflective of real world data collection. In this section, we address how *site-based* representative sampling strategies impact the relationship between group fairness and data representativeness. To begin, we design site-based representative sampling algorithms and apply the most effective such algorithms to construct datasets and study the impact of representativeness targets on group fairness. Sites are induced through the location variables outlined in table 1. We exclude any sites with fewer than 2,000 records to prevent sites with few records, which may represent an unstable or incomplete data distribution, from being selected.

### 6.1    Constructing Representative Datasets

We begin by examining the feasibility of constructing representative datasets when the data collector lacks direct access to the underlying-dataset. For each domain, we define groups over the joint distribution of sensitive features outlined in table 1. For example, on the Law School data domain, groups are defined as combinations of age, gender, and race. Following Celis et al. (2020), we set the target vector of each domain to $\mathbf{v} = \langle 0.5, \cdots, 0.5 \rangle$. Each algorithm is run over 50 timesteps, selecting 40 samples at each step, to yield 2,000 total samples. In Figure 2 we present results for dataset construction in the Law School domain and measure representativeness via mean squared error (MSE, $\mathcal{M}$ in section 4.3). Lower MSE indicates a more representative dataset. Naïve algorithms such as $\varepsilon$-Greedy and RND have notably worse performance compared to more sophisticated algorithms such as PBRS and UCB-BY. Distributed sampling is more effective than non-distributed sampling, as seen by the difference between PBRS and D-PBRS. We also observe that all algorithms, including the myopic optimal (OPT) and distributed myopic optimal (D-OPT), have a nontrivial MSE. This indicates that collecting perfectly representative datasets may not be feasible in practice and some degree of unrepresentativeness is unavoidable.

### 6.2    Site-Based Sampling and Fairness

Next, we examine how sampling strategies impact the relationship between fairness and representativeness. In these experiments, we consider one binary sensitive feature (e.g. race) at a time; again, we set $G_0$ to be the underrepresented group in the underlying-dataset. To construct datasets with varying representativeness

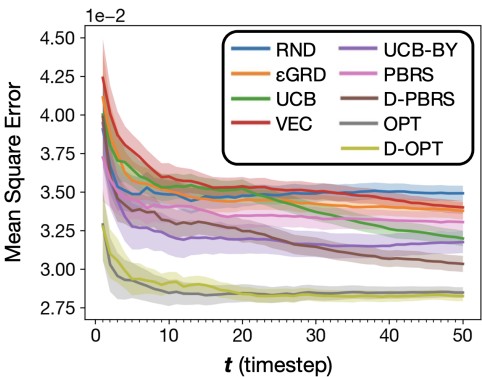

Figure 2: Site-based dataset construction with various algorithms in the Law School data domain. The y-axis measures mean squared error from the cohort attribute distribution to the target attribute distribution and, thus, is inversely correlated with representativeness. D-PBRS generates the most representative cohort of all non-fully-informed sampling algorithms.

via site-based sampling strategies, we set the target vector to take on values $v = 0, 0.05, \ldots, 0.95, 1$. We find that site-based sampling substantially impacts the relationship between group representation and fairness. In figure 3, datasets collected through SRS, a non-site-based technique, show better fairness (overlapping $G_0$ and $G_1$ AUC confidence intervals) as the fraction of $G_0$ in the training data increases. Site-based sampling shows a different trend: oversampling either group ($G_0$ proportions further from the black dotted line) causes substantial AUC drops for both groups and no clear fairness trend. For the same level of representativeness, subgroup and population AUCs achieved by site-based sampling are noticeably lower. Thus, when building intuition solely from direct-sampling methods, attempting to achieve a desired group representation through site-based sampling may yield unexpected downstream results. Specific site-based sampling strategies differ slightly; for instance, the distributed strategies D-PBRS and D-OPT support a wider range of $G_0$ proportions before seeing a significant AUC drop-off. Nevertheless, all site-based sampling strategies share qualitative similarities in the representativeness-fairness relationship that distinguish them from SRS.

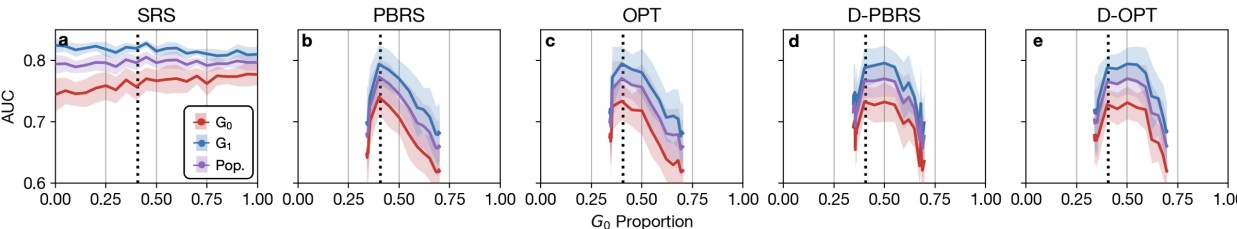

Figure 3: Sampling from sites affects the relationship between representation and fairness but the specific sampling method has less impact. We show population (purple) and subgroup (red and blue) AUCs with 95% CIs for GBC models in the Texas Salary data domain with "Race: White" as the sensitive feature of interest. Datasets are built using one of five representative sampling methods: stratified random sampling (SRS; **a**), prior-based representative sampling (PBRS; **b**), fully-informed optimal sampling (OPT; **c**), distributed prior-based representative sampling (D-PBRS; **d**), and distributed fully-informed optimal sampling (D-OPT; **e**). The dotted black line indicates the dataset group $G_0$ proportion. Site-based samplers are restricted to the $G_0$ proportions available at sites, so they cannot generate as wide a proportion range as SRS.

## 7 Fair-Sampling Algorithms and Representativeness

We now examine the complementary question of how sampling for fairness impacts dataset representativeness. Unlike representative sampling algorithms, fair sampling algorithms do not target a particular group balance. Instead, a group balance emerges secondarily as a result of selecting records from the sensitive feature group

($G_0$ or $G_1$) with lower performance. This approach is often called minimax fairness and it follows the recommendations from Chen et al. (2018) to oversample the group with worse model performance. There have been several approaches proposed for fair record selection (Abernethy et al., 2020; Shekhar et al., 2021), of which we adapt the former to successively identify the worse-performing group in the training partition of each data domain and draw 40-example mini-batches from said group, a process we term fair direct sampling. We achieve fair arm-based sampling (Appendix A.3) by modifying fair sampling to select the single site with the highest proportion of the worse-performing group. Both fair arm-based and fair direct sampling, are repeated 21 times in each train/test fold.

| Feature | Fair Direct Sampling | Fair Site-Based Sampling | Census Proportions |
|---|---|---|---|
| **Law School** | | | |
| Race | **.803**$_{\pm.267}$ | **.197**$_{\pm.017}$ | .38 |
| Age | .585$_{\pm.378}$ | **.359**$_{\pm.011}$ | .26 |
| Gender | .299$_{\pm.280}$ | **.443**$_{\pm.013}$ | .50 |
| **Intensive Care** | | | |
| Ethnicity | .260$_{\pm.330}$ | **.272**$_{\pm.034}$ | .38 |
| Age | .488$_{\pm.296}$ | .520$_{\pm.031}$ | .82 |
| Gender | .720$_{\pm.299}$ | **.442**$_{\pm.017}$ | .50 |
| **Lending Club** | | | |
| Mortgage | .431$_{\pm.247}$ | .502$_{\pm.078}$ | .40 |
| Rent | .475$_{\pm.263}$ | **.357**$_{\pm.100}$ | .33 |
| Own | **.789**$_{\pm.229}$ | **.168**$_{\pm.026}$ | .25 |
| **Texas Salary** | | | |
| Race: Black | **.007**$_{\pm.011}$ | **.047**$_{\pm.032}$ | .12 |
| Race: White | **.936**$_{\pm.145}$ | **.696**$_{\pm.097}$ | .50 |
| Race: Other | **.874**$_{\pm.187}$ | **.417**$_{\pm.044}$ | .38 |
| Sex | .497$_{\pm.409}$ | .550$_{\pm.268}$ | .50 |

Table 2: The $G_0$ proportions in training data acquired by fair sampling strategies, and the proportions of $G_0$ reported by the U.S. Census Bureau (U.S. Census Bureau, 2024) Table P1 for demographics and Organisation for Economic Co-Operation and Development (OECD) for home ownership (OECD, 2022). Bold, and underlined values indicate group proportions that are more than one standard deviation away for equally balanced groups (i.e., $\mathbf{v} = 0.5$) and census proportions, respectively.

Table 2 shows the group $G_0$ (data domain minority group) proportions induced by fair sampling. This table also shows the proportions which $G_0$ in the US census (U.S. Census Bureau, 2024). When unfairness exists (Appendix C.3), fair sampling methods prioritize the worse performing group (often, but not always, $G_0$). However, this prioritization can lead to heavily imbalanced datasets (e.g., Law School with Race, and Texas Salary with Race: White). Additionally, fair sampling frequently results in datasets where the proportions of $G_0$ are neither close to balanced (bold) nor close to census proportions (underlined). This can also result in datasets where $G_0$ is underrepresented (e.g., Texas Salary with Race: Black and Intensive Care with Age). These results are complementary to observations on representative sampling algorithms; fair sampling algorithms can frequently result in unrepresentative datasets, and representative sampling algorithms can frequently result in datasets with high downstream unfairness.

## 8 Effects of Model Complexity

As noted in Theorem 1, group representation is only one component of unfairness; we hypothesize that differential learnability between groups is also a diving factor in the relationship between representativeness and fairness. Learnability closely aligns with what Chen et al. (2018) termed the model-dependent bias term and model-independent noise term. Building upon their work and our theoretic results, we hypothesize that a more complex model can better capture complex relationships between features and labels. If our hypothesis is correct, increasing model capacity should decrease unfairness between groups *independently of*

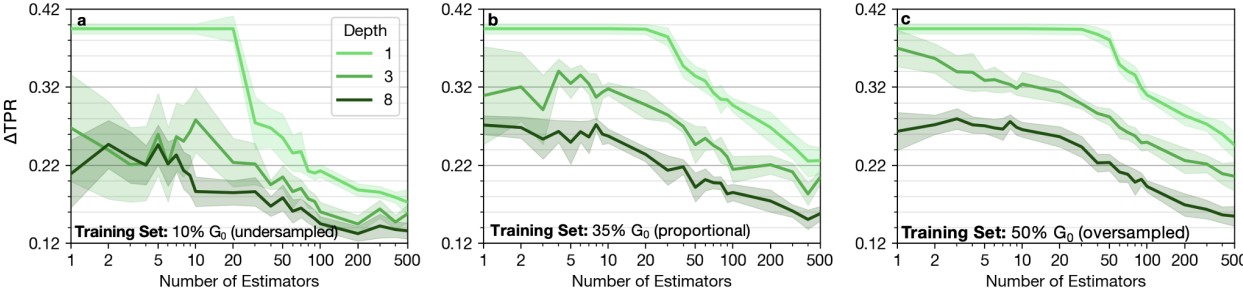

Figure 4: True Positive Rate unfairness ($\Delta$TPR) between groups (defined by gender) as a function of model complexity (number of estimators and tree depths) for three different levels of representativeness of $G_0$: 10% (left), 35% (center) and 50% (right) of the training data. Roughly 35% of the underlying distribution is from group $G_0$. Panel **a** demonstrates magnified underrepresentation, panel **b** demonstrates dataset representation, and panel **c** demonstrates both class balance and population representation. Shaded regions indicate 95% CIs.

*group representation.* Thus, we would expect to see an improvement in AUC, TPR, and TNR parity when using higher capacity models.

To control complexity, we test simultaneous combinations of GBC hyperparameters of `max_tree_depth`[4] from 1 to 8 and `n_estimators` from 1 to 500. As both hyperparameters increase, model capability increases. As a parallel analysis, we measure model performance alongside our fairness analysis. Certain fairness-improving measures are known to harm overall model performance (e.g., overall AUC), often termed the accuracy-fairness trade-off. Identifying a decrease in model performance with increasing model complexity would not invalidate a putative improvement in fairness, but it may provide additional context for applying these techniques.

We show how increasing complexity through greater tree depth and more estimation steps can reduce TPR unfairness ($\Delta$ TPR) between gender groups (Fig. 4). We find that increased fairness from model complexity occurs independent of the training dataset composition by testing three cases: heightened underrepresentation (10% $G_0$), near-test-set proportion (35% $G_0$), and balanced/population representation (50% $G_0$). This trend holds similarly for other datasets and fairness metrics (AUC and TNR parity), but TPR parity appears to benefit the most from increased complexity, in general (appendix C.4). Lastly, we observe that classifier accuracy tends to peak at intermediate complexity levels. The improvements in both accuracy and fairness from low to intermediate complexity are likely due to a more capable model being able to capture complex data relationships.

# 9 Discussion

Representative datasets yield several benefits, including legitimacy, validity, equity, and generalizability. Generalizability is closely related to algorithmic fairness, a measure of prediction or performance parity between different groups. In this paper, we analyze the relationship between representation and downstream algorithmic fairness in classification, finding that more representative datasets rarely yield fairer classifiers. Complementarily, datasets constructed to promote algorithmic fairness are rarely representative of the overall population. In practice, this tension between representativeness and algorithmic fairness means that data collectors must, at times, prioritize one property over the other. To explain this tension, we theorize that representativeness and fairness are at odds when groups differ substantially in their difficulty to learn. If a large difficulty gap exists between groups, adding data points from the more difficult group may not be sufficient to overcome the disparity in classifier performance. Alternatively, we show that increasing model complexity can sometimes help close this performance gap.

---

[4]The capability of individual decision trees to capture complex relationships is driven primarily by the number of internal nodes (max depth) (Leboeuf et al., 2020; Bentéjac et al., 2021).

Furthermore, we also find that the messy nature of realistic data collection complicates the relationship between representativeness and fairness. As seen in figure 3a, subgroup representation contributes to unfairness because increasing the proportion of group $G_0$ ameliorates the AUC disparity in this idealized example. Yet, attempting to increase the proportion of group $G_0$ through site-based sampling causes a precipitous performance decrease for all groups. Site-based sampling consequentially affects the distribution of features, groups, and labels, resulting in notably different classifier performance compared with direct sampling. As such, it is important to consider *how* a dataset is constructed beyond its demographics matching a target distribution. Thus, we nuance the suggestions of prior works like Chen et al. (2018), which recommend oversampling the group with lower performance, to ensure that such oversampling does not have undesired effects.

Despite the contributions of this work, there are some key limitations to note. Representative sampling, as we have formulated it, focuses on matching attribute means to a target population; however, the underlying distributions of the dataset and target population may differ substantially. While matching attribute means may be intuitive for physical or biological variables like age, it becomes much more complicated for social variables like race, where the notion of ground truth does not necessarily apply. Finally, it is important to note that increasing model complexity may not always be able to substantially improve algorithmic fairness (as theorized by Chen et al. (2018)). Future work may include broader definitions of representation that are not group-centric, as well as expanding these results to additional definitions of fairness like procedural fairness as opposed to classifier parity measures.

We conclude that the relationship between dataset representativeness and downstream fairness is complicated and influenced by numerous factors. While increasing a group's representation in a dataset sometimes improves that group's performance substantially, the practical constraints of dataset generation may sometimes cause the opposite effect. Alternatively, changing a group's representation in a dataset can have a negligible impact on classifier performance; as shown, this may be due to learnability differences between groups.

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

# A  Sampling

## A.1  Preliminaries

Let $D$ be a distribution over $\mathcal{X} \times \mathcal{A} \times \mathcal{Y}$, i.e., $D$ is the *true* population distribution. The data collector does not know the distribution $D$, but may know its mean. Let $\mathcal{S} = \{S_1, \ldots, S_m\}$ be the set of $m$ *sites*,

where each site $S_j$ is associated with an underlying site-specific population distribution $D_j$ over $\mathcal{X} \times \mathcal{A} \times \mathcal{Y}$. Importantly, the distribution $D_j$ for every site $j \in [m]$ is unknown to the data collector. Over the course of $T$ timesteps, the data collector will sequentially recruit samples from sites $\mathcal{S}$, with the objective of building a *representative* final dataset. Each sample from site $S_j^{(t)}$ constitutes a draw $(\boldsymbol{X}, \boldsymbol{A}, \boldsymbol{Y})^{(t)} \sim D_j$. After $T$ rounds, the data collector has a dataset $(\boldsymbol{X}, \boldsymbol{A}, \boldsymbol{Y}) = \bigcup_{t=1}^{T}(\boldsymbol{X}, \boldsymbol{A}, \boldsymbol{Y})^{(t)}$. Given a *target* demographic vector $\boldsymbol{v} \in \mathbb{R}^d$, which represents the ideal mean of $\boldsymbol{A}^{(T)}$, the data collector aims to sample such that $\mathrm{avg}(\boldsymbol{A}^{(T)})$ is as close to $\boldsymbol{v}$ as possible. Thus, we conceptualize representativeness as inversely proportional to the distance from $\mathrm{avg}(\boldsymbol{A}^{(T)})$ to $\boldsymbol{v}$; as this distance decreases, representativeness increases. For example, suppose there are two binary features of interest: gender (Male or Female) and age (Young or Old). A target vector of $\boldsymbol{v} = \langle 0.3, 0.7 \rangle$ implies that an ideal dataset is 30% Male and 70% Young. Therefore if $\mathcal{M}$ is the $\ell_1$-norm then a dataset which is 25% Male and 60% Young would be 0.15-distant with respect to $\boldsymbol{v}$. We next formally define representativeness.

Given target vector $\boldsymbol{v}$, the objective of sampling the most representative dataset can be expressed as

$$\min_{(\boldsymbol{X}, \boldsymbol{A}, \boldsymbol{Y})} \mathcal{M}\left(\boldsymbol{v}, \frac{1}{|\boldsymbol{A}|}\sum_{\boldsymbol{a} \in \boldsymbol{A}}\boldsymbol{a}\right) \mathrm{s.t.}(\boldsymbol{X}, \boldsymbol{A}, \boldsymbol{Y}) = \bigcup_{t=1}^{T}(\boldsymbol{X}, \boldsymbol{A}, \boldsymbol{Y})^{(t)}. \tag{1}$$

We limit $\mathcal{M}$ to distance measures which are convex in the collected set of sensitive features $\boldsymbol{A}$, including all $\ell_p$-norms with $p \geq 1$ and KL-divergence. It should be recognized that a key challenge with representative sampling is that the objective in Problem 1 is not supermodular, even for convex $\mathcal{M}$, as a function of $\frac{1}{|\boldsymbol{A}|}\sum_{\boldsymbol{a} \in \boldsymbol{A}}\boldsymbol{a}$. This is due to the nonlinear nature of the average $\frac{1}{|\boldsymbol{A}|}\sum_{\boldsymbol{a} \in \boldsymbol{A}}\boldsymbol{a}$, with respect to samples.

## A.2 Convex Formulation and Prior-Based Sampling

In this section, we first demonstrate how the data collector's sampling problem can be formulated through the framework of multi-armed bandit with concave reward (convex loss in our case). Utilizing this particular problem structure, we present our algorithm for constructing representative datasets. Our strategy for optimizing this objective is to provide a modified form of the objective in Equation 1 which is convex with respect to the samples collected at each time step. To do this, we first note that each iteration returns $k$ data points[5], and thus the final dataset will consist of $Tk$ examples, and the average demographic vector of the dataset can be written as

$$\frac{1}{|\boldsymbol{A}|}\sum_{\boldsymbol{a} \in \boldsymbol{A}}\boldsymbol{a} = \frac{1}{T}\sum_{t=1}^{T}\sum_{\boldsymbol{a} \in \boldsymbol{A}^{(t)}}\frac{\boldsymbol{a}}{k} = \frac{1}{T}\sum_{t=1}^{T}\mathrm{avg}(\boldsymbol{A}^{(t)}) \tag{2}$$

where $\mathrm{avg}(\boldsymbol{A}^{(t)})$ is the average demographic vector present in the sample $\boldsymbol{A}^{(t)}$ collected at time $t$.

With this fact, the data collector's objective can be expressed as a function simply of the sum of the means from each sample,

$$\min_{\boldsymbol{A}} \mathcal{M}\left(\boldsymbol{v}, \frac{1}{T}\sum_{t=1}^{T}\mathrm{avg}(\boldsymbol{A}^{(t)})\right) \tag{3}$$

**Theorem 4.** *The objective in Equation 3 is convex with respect to the sample values $\mathrm{avg}(\boldsymbol{A}^{(t)})$ and has an equivalent optimal value with Equation 1 after all $T$ rounds are completed.*

*Proof.* The objective in Equation 1 is

$$\min_{(\mathbf{X}, \mathbf{A}, \mathbf{Y})} \mathcal{M}\left(\mathbf{v}, \frac{1}{|\mathbf{A}|}\sum_{\mathbf{a} \in \mathbf{A}}\mathbf{a}\right)$$

---

[5]The convex formulation holds when, in expectation, each iteration yields $k$ data points.

and the objective in Equation 3 is

$$\min_{\mathbf{A}} \ \mathcal{M}\left(\mathbf{v}, \ \frac{1}{T}\sum_{t=1}^{T} \mathrm{avg}\big(\mathbf{A}^{(t)}\big)\right)$$

To first prove equivalence between these two objectives when each sample yields $k$ individuals, we restate the derivation provided in the main

$$\frac{1}{|\mathbf{A}|}\sum_{\mathbf{a}\in\mathbf{A}}\mathbf{a} = \frac{1}{Tk}\sum_{\mathbf{a}\in\mathbf{A}}\mathbf{a} = \frac{1}{T}\sum_{t=1}^{T}\sum_{\mathbf{a}\in\mathbf{A}^{(t)}}\frac{\mathbf{a}}{k} = \frac{1}{T}\sum_{t=1}^{T}\mathrm{avg}\big(\mathbf{A}^{(t)}\big)$$

as such, we see that for any $\mathbf{A} = \cup_{t=1}^{T}\mathbf{A}^{(t)}$,

$$\mathcal{M}\big(\mathbf{v}, \frac{1}{|\mathbf{A}|}\sum_{\mathbf{a}\in\mathbf{A}}\mathbf{a}\big) = \mathcal{M}(\mathbf{v}, \frac{1}{T}\sum_{t=1}^{T}\mathrm{avg}\big(\mathbf{A}^{(t)}\big)),$$

and the two objectives have equal optimums.

To show the convexity of the data collector's objective w.r.t. the samples $\mathbf{A}^{(t)}$, we note that $\mathcal{M}(\mathbf{v}, \mathbf{u})$, is convex in $\mathbf{u} \in \mathbb{R}^d$, and thus for any linear function $f$, the composition $\mathcal{M}\big(\mathbf{v}, f(\mathbf{u})\big)$ is also convex in $\mathbf{u}$. The function $\frac{1}{T}\sum_{t=1}^{T}\mathrm{avg}\big(\mathbf{A}^{(t)}\big)$ is linear in the collection of samples $\mathbf{A}^{(T)} = \cup_{t=1}^{(T)}\mathbf{A}^{(t)}$. Thus, $\mathcal{M}$ is convex in the samples $\mathbf{A}^{(T)}$. $\qquad\square$

Since the samples returned by each site at time $t$ can now be thought of as a single vector $\mathrm{avg}\big(\boldsymbol{A}^{(t)}\big)$, and the loss function $\mathcal{M}$ is convex with respect to those sample vectors, the problem of representative sampling can be naturally formulated as a multi-armed bandit problem with convex loss. We next discuss Bayesian sampling procedure which can capitalize on both this convex formulation as well as site-wise prior information.

## A.3 Algorithms

### A.3.1 Prior-based Bayesian Representative Sampling (PBRS)

Before outlining the details of our algorithm, we first discuss the motivation behind PBRS (Alg. 1), which is twofold. First, in many real-world domains where representativeness is a salient issue, a wealth of summary data is available, which allows data collectors to form reasonably accurate priors over the distributions at each site. Second, the Bayesian nature of our approach always for dynamic control over how aggressively the prior distributions are updated after each sample, this is particularly useful in settings where the distributions at sites may change over time (a common occurrence in the real-world), such shifts are discussed in Section A.4.1.

PBRS works by maintaining an estimate of the distribution of groups at each site $D_j'$, which corresponds to a multinomial distribution, when sensitive features are binary and a multivariate-normal distribution when sensitive features are continuous. In the former $D_j' = M_d(k, p_{j,1}, \ldots, p_{j,d})$, where $p_\ell$ gives the probability that an individual sampled from site $j$ will have sensitive feature $\ell$ equal to 1. In the latter, $D_j' = \mathcal{N}(\boldsymbol{\theta}_j', \boldsymbol{\Sigma}_j')$ where $\boldsymbol{\theta}_j'$ and $\boldsymbol{\Sigma}_j'$ are the mean and covariance of sensitive features at site $j$. In both cases, each distribution is initialized via a prior estimate of the true distribution at site $j$. In the case that no prior is provided, a *default* prior can be induced by either assigning uniform values to each parameter (e.g., $\boldsymbol{\theta}_j' = \mathbf{0}$ and $\boldsymbol{\Sigma}_j' = I_d$), or as values from the target vector $\boldsymbol{v}$ (e.g., $p_{j,\ell} = \boldsymbol{v}[\ell]$ for all $\ell \in [d]$). Throughout the course of constructing the dataset, the samples obtained at each time step can be used to update these distributions to more accurately reflect the true distribution of each site. To do this, we use the conjugate prior of each distribution to iterative update the estimation $D_j'$. In the case of binary group features, the conjugate prior is represented by a Dirichlet distribution $\mathrm{Dir}(d, \alpha_{j,1}, \ldots, \alpha_{j,d})$, and in the case of continuous group features, the conjugate prior is represented by an inverse Wishart distribution $\mathcal{W}_j^{-1}(\boldsymbol{\theta}_j', \boldsymbol{\Psi}_j, n_j)$.

At each time step $t$, the estimated distribution $D_j'$ is induced by sampling parameters from the corresponding conjugate prior, and is then used to compute the expected improvement to $\mathcal{M}\big(\boldsymbol{v}, \mathrm{avg}(\boldsymbol{A})\big)$ for each site.

---

**Algorithm 1:** Prior-based Bayesian Representative Sampling (PBRS) and Distributed PBRS (D-PBRS): sampling procedures for building a representative dataset.

---

**Data:** sites $\mathcal{S}$, budget $T$, target vector $\boldsymbol{v}$, prior mean and covariance $\boldsymbol{\theta}_j, \boldsymbol{\Sigma}_j \forall j \in [m]$.
**Result:** dataset $(\boldsymbol{X}, \boldsymbol{A}, \boldsymbol{Y})$.

1   $n_j \leftarrow 0 \quad \forall j \in [m]$;                                             // Number of times site $j$ is sampled

2   $\boldsymbol{\Psi_j} = (n_j + 1)\hat{\boldsymbol{\Sigma}}_j$;                        // Inverse scale matrix of normal inverse Wishart

3   $\mathcal{W}_j^{-1}(\boldsymbol{\theta}_j, \boldsymbol{\Psi_j}, n_j)$;                 // Initialize normal inverse Wishart distribution

4   $(\boldsymbol{X}, \boldsymbol{A}, \boldsymbol{Y}) = \emptyset$;                                                // Initialize dataset

5   **for** $t = 0 \ldots T$ **do**

6      $\hat{\boldsymbol{\theta}}_j, \hat{\boldsymbol{\Sigma}}_j \sim \mathcal{W}_j^{-1}(\boldsymbol{\theta}_j, \boldsymbol{\Psi}_j, n_j), \ \forall j$;

7      $\hat{\boldsymbol{a}}_j \sim \mathcal{N}(\hat{\boldsymbol{\theta}}_j, \hat{\boldsymbol{\Sigma}}_j), \ \forall j$;

8      $(\boldsymbol{X}, \boldsymbol{A}, \boldsymbol{Y})$.add$\big($allocateAndSample$(\boldsymbol{A}, \boldsymbol{v}, \boldsymbol{a}, T, n)\big)$;

9      updatePriors$\big(\boldsymbol{A}^{(t)}, \boldsymbol{\theta}_{j^*}, \boldsymbol{\Psi}_{j^*}, n_{j^*}, \beta, t\big)$;

10 **end**

11 **return** $\bigcup_{t=1}^{T}(\boldsymbol{X}, \boldsymbol{A}, \boldsymbol{Y})^{(t)}$;                                     // Final dataset

12 **Function** allocateAndSample$(\boldsymbol{A}, \boldsymbol{v}, \boldsymbol{a}, T, n)$:

13      // PBRS

14      $j^* \leftarrow \arg\min_j \mathbb{E}\left[\mathcal{M}\left(\boldsymbol{v}, \big(\text{sum}(\boldsymbol{A}) + \boldsymbol{a}_j\big)/T\right)\right]$;

15      // Arm with best improvement

16      $(\boldsymbol{X}, \boldsymbol{A}, \boldsymbol{Y})^{(t)} \sim D_{j^*}$;                       // Sample data from arm $j^*$

17      $n_{j^*} += 1$;

18      **return** $(\boldsymbol{X}, \boldsymbol{A}, \boldsymbol{Y})^{(t)}$;

19 **Function** allocateAndSample$(\boldsymbol{A}, \boldsymbol{v}, \boldsymbol{a}, T, n)$:

20      // Distributed PBRS

21      $\rho_j \leftarrow 0 \quad \forall j \in [m]$                                      // Resource vector

22      $\rho^* \leftarrow \arg\min_\rho \mathbb{E}\left[\mathcal{M}\left(\boldsymbol{v}, \ \big(\text{sum}(\boldsymbol{A}) + \rho\boldsymbol{a}\big)/T\right)\right]$

23        subject to $\Sigma\rho = 1$

24      $(\boldsymbol{X}, \boldsymbol{A}, \boldsymbol{Y})^{(t)} = \emptyset$

25      **for** $j = 0 \ldots m$ **do**

26        $(\boldsymbol{X}, \boldsymbol{A}, \boldsymbol{Y})^{(t,j)} \sim \lfloor \rho_j^* \rfloor D_j$;     /* Sample from arm j a fraction of examples determined by $\rho^*$ for $j$ */

27        $(\boldsymbol{X}, \boldsymbol{A}, \boldsymbol{Y})^{(t)}$.add$\big((\boldsymbol{X}, \boldsymbol{A}, \boldsymbol{Y})^{(t,j)}\big)$;

28        $n_j += \rho_j^*$;

29      **end**

30      **return** $(\boldsymbol{X}, \boldsymbol{A}, \boldsymbol{Y})^{(t)}$;

---

**Algorithm 2:** Fair Arm-Based Sampling

---

**Data:** Sites $\mathcal{S}$, classifier $\mathcal{F}$, loss function $\mathscr{L}(\mathcal{F}(\boldsymbol{X}), \boldsymbol{Y})$
**Result:** dataset $(\mathbf{X}, \mathbf{A}, \mathbf{Y})$

1 randomly sample initial data $(\mathbf{X}, \mathbf{A}, \mathbf{Y})$;

2 **for** $t = 1 \ldots T$ **do**

3      train $\mathcal{F}$ using current data $(\mathbf{X}, \mathbf{A}, \mathbf{Y})$;

4      $g^* \leftarrow$ group with with the highest loss w.r.t, $\mathcal{F}$, and $\mathscr{L}$;

5      $s_j^* \leftarrow$ site with the largest expected proportion of $g^*$;

6      sample new data $(\boldsymbol{X}^{(t)}, \boldsymbol{A}^{(t)}, \boldsymbol{Y}^{(t)})$ from site $s_j^*$ update dataset $(\mathbf{X}, \mathbf{A}, \mathbf{Y}) \cup = (\mathbf{X}^{(t)}, \mathbf{A}^{(t)}, \mathbf{Y}^{(t)})$;

7 **end**

8 **return** $(\boldsymbol{X}, \boldsymbol{A}, \boldsymbol{Y})^{(t)}$;

---

**Algorithm 3:** Bayesian UCB (UCB-BY)

---

**Data:** Sites $\mathcal{S}$, prior distributions $D$, target vector $\mathbf{v}$, metric $\mathcal{M}$
**Result:** dataset $(\mathbf{X}, \mathbf{A}, \mathbf{Y})$

**1** $U, L \leftarrow$ initialized bounds based on $D$
**2** **for** $t = 1 \ldots T$ **do**
**3** $\quad$ $s_j^* \leftarrow$ site with highest representativeness based on $L$
**4** $\quad$ sample new data $(\boldsymbol{X}^{(t)}, \boldsymbol{A}^{(t)}, \boldsymbol{Y}^{(t)})$ from site $s_j^*$ update dataset $(\mathbf{X}, \mathbf{A}, \mathbf{Y}) \cup = (\mathbf{X}^{(t)}, \mathbf{A}^{(t)}, \mathbf{Y}^{(t)})$;
**5** $\quad$ update distributions $D$
**6** $\quad$ update bounds $U, L$ based on $D$.
**7** **end**
**8** **return** $(\boldsymbol{X}, \boldsymbol{A}, \boldsymbol{Y})^{(t)}$;

---

PBRS selects the site $j^*$, corresponding to the maximum expected improvement. The sample from site $j^*$ is then used to update conjugate prior. To better anticipate the possibility for site bias, we incorporate a hyperparameter $\beta \geq 1$ which modifies the procedure through which conjugate distributions are updated by increasing the strength of samples from minority groups by a factor of roughly $\beta^{(1-t/T)}$. This hyperparameter incentivizes PBRS to more aggressively search for sites which yield individuals from minority groups, thus helping to circumvent site bias towards those groups.

### A.3.2 Distributed Prior-based Bayesian Representative Sampling (D-PBRS)

D-PBRS (Alg. 1) modifies PBRS to allow multiple sites to be sampled from simultaneously in a single timestep, still limited to $k$ total samples per timestep. D-PBRS distributes the budget $k$ according to a vector $\rho$, which is selected to maximally decrease $\mathcal{M}$ given all previously collected samples, with the constraint that $\Sigma\rho = 1$. In the sampling step, $k$ total samples are divided among the sites according to $\rho$ with fractional sample allocations rounded down, and assigned to the site that minimizes $\mathcal{M}$. For example, int he case of two sites and a budget of $k = 40$, $\rho = \langle .75, .25 \rangle$ implies collecting 30 samples from the first site, and 10 from the second site.

### A.3.3 Fair Arm-Based Sampling

We introduce a third arm sampling procedure (Alg. 2), one designed to optimally improve minmax algorithmic fairness. We enact this goal by first training a classifier on the available dataset, then evaluating its group-specific performance on a set of validation data. Next, we identify the group with the lowest AUC and sample from the arm with the highest proportion of that group. This algorithm represents an adaptation of previous work by Abernethy et al. (2020) and Shekhar et al. (2021) to our arm-based selection process.

### A.3.4 Bayesian UCB

Finally, we introduce a fourth arm-based sampling procedure which modifies the existing UCB algorithm with a Bayesian framework (UCB-BY, Alg. 3). UCB-BY isolates the benefit of the Bayesian framework on top of an existing and well-known multi-armed bandit algorithm.

### A.4 Sampling Procedure and Algorithms

For a target demographic vector $\boldsymbol{v}$ and a distance measure $\mathcal{M}$, we iteratively select a site (or mix of sites) and receive $k$ data points $(\mathbf{x}, \mathbf{a}, \mathbf{y})$ randomly sampled from the partition corresponding to that site. After repeating the process $T$ times, we combine the $T \cdot k$ data points into a single dataset and compute the distance between the target demographic vector and the average demographics of the constructed dataset $\mathcal{M}(\boldsymbol{v}, \text{avg}(\boldsymbol{A}))$. To demonstrate the improved efficacy of **PBRS** (BY(H) and BY(L) for high- and low-noise priors) and **D-PBRS** (DS(H) and DS(L) for high- and low-noise priors), we compare to three baselines: 1) $\varepsilon$-**Greedy** ($\varepsilon$GRD): which randomly selects a site with probability $\varepsilon$ and otherwise selects the site which has the maximum expected decrease in error; 2) **UCB-LCB** Agrawal & Devanur (2014) (UCB): which is a UCB-based algorithm Auer et al. (2002) for solving multi-armed bandit problems with convex loss; and 3)

**OL-Vec** Kesselheim & Singla (2020) (VEC), which derives a one dimensional function to approximate the distance measure $\mathcal{M}$ and uses online convex minimization to select the site at each timestep. In addition to the aforementioned baselines, we also compared to random site selection **Random** (RND), and **OPT**, a policy that has full information and selects the site corresponding to the maximum expected decrease in error. This baseline serves as the best possible *myopic* sampling scheme when the data collector is limited to a single site per timestep. To test our representative sampling algorithms, we analyzed a setting in which there are 20 arms (achieved via either randomly subsampling or duplicating sites, depending on the number of sites in the dataset), 50 time steps, and sample sizes of $k = 40$ individuals. Based on this setup, the constructed dataset corresponds to 2,000 examples. We use a class-balanced target vector, $\boldsymbol{v} = \langle .5, \cdots, .5 \rangle$, and average performance across 100 experiments. To measure how effective each sampling algorithm is at producing a representative dataset with respect to a target demographic vector $\boldsymbol{v}$, we use the $\ell_2$-norm, $\mathcal{M}\big(\boldsymbol{v}, \mathrm{avg}(\boldsymbol{A})\big) = \|\boldsymbol{v} - \mathrm{avg}(\boldsymbol{A})\|)$.

### A.4.1 Site Variations

**No bias** is our baseline. In this setting, site response distributions are induced by the location-based partitions and do not change over time.

**Response bias** occurs when certain demographics appear at sites with disproportionately high (or low) frequencies compared to other groups. For example, as shown in Association (2021) the ratio of individuals identifying as ethnic minorities is substantially lower at the majority of law schools compared to the population. Response bias can be modeled using coefficients $\lambda \in \mathbb{R}_{\geq 0}$ and $\gamma \leq m$, where members of majority groups are $\lambda$-times more likely to respond at $\gamma$ sites compared to their base response rate at those sites. For example suppose there is one binary feature (i.e., two groups), $\gamma = m/2$, and $\lambda = 4$, then individuals from the majority group are 4-times more likely to appear in a sample from half of the sites. The no variation setting is recovered when $\lambda = 1$. We evaluate the representativeness of the final datasets constructed by the tested algorithms across a range of $\lambda$ from 0.1 to 10. We can convert $\lambda$ to a proportion scaling factor $b$ through the transformation $b = \frac{\lambda}{1+\lambda}$. To implement response bias for binary sensitive features $\mathbf{A} = \{0, 1\}^d$, we choose 1 to represent the larger group. For example if there are two features, age (Old or Young) and gender (Male or Female), where 70% of individuals are Old and 60% are Female, then $\langle 1, 1 \rangle$ corresponds to an individual who is both Old and Female. When sampling from site $j$, rather than selecting $k$ examples uniformly at random from the associated data partition, $k$ examples are selected randomly with weights proportional to $\sum_{\ell=1}^{d} \big(b \cdot a_\ell + (1-b) \cdot (1-a_\ell)\big)^2$. Thus an individual with features in each majority group (i.e., $\mathbf{a} = \mathbf{1}$) has $d \times \lambda^2$ times more sample weight than an individual with features from each minority group (i.e., $\mathbf{a} = \mathbf{0}$). When $\lambda = 1$, then $b = 0.5$ and this sum reduces to $0.5^2$ for *all* individuals and the no-bias setting is recovered.

Lastly, **causal distribution shifts** occur when demographic distributions at each site change over time as the result of the data collector's decisions. When selection is desirable (e.g., monetary compensation for participating in trials), individuals may modify their behavior in order to be selected. Causal distribution shifts affect response probability $p$ of each individual at site $j$ with coefficient $\alpha \in \mathbb{R}_{\geq 0}$ s.t. $p_{post} = p_{pre}^{1+\alpha \times \rho_j}$. We evaluate the representativeness of the final datasets constructed by the tested algorithms across a range of $\alpha$ from 0.1 to 10 using $\lambda = 2$ such that there is a response bias to causally magnify. To implement this for binary groups, we again represent each majority group with value 1 and minority groups with value 0. Similar to the case of response bias, we re-weight the sample probabilities of the data partition associated with each site. The sample probabilities for each individual at site $j$, after $n_j$ sampling iterations, is proportional to $\prod^{\sum^{n_j} \rho_j} p^{1+\alpha \times \rho_j}$, where $p$ is the initial response probability of the individual, determined as described in the response bias section above. As $\alpha$ or $\lambda$ increase, members from minority groups are less likely to appear in repeated samples from the same site.

## B  Data Preprocessing and Computing Infrastructure

For each dataset we follow a uniform procedure when preprocessing the raw data files. For datasets 1-4, we partition the dataset into $m$ disjoint sets sharing the same location, inducing $m$ sites (i.e., arms).

The Law School, Intensive Care, and Texas Salary datasets include location information corresponding to *actual* sites, such as the student's law school. For the Lending Club dataset, we induce sites by U.S. state. The Adult Income and Community Crime datasets do not have applicable location information, so they are not used to evaluate our sampling algorithms. Nevertheless, these two datasets have well-documented algorithmic fairness limitations, making them ideal case studies for our fairness analyses. Sites with fewer than 2,000 records were excluded from analysis due to small sample size limitations. Ordinal features (e.g., an individual's income) are scaled between 0 and 1. Non-ordinal categorical features (e.g., an individual's occupation) are one-hot encoded. Binary features (e.g., ) are encoded as 0 and 1. All sensitive features are treated as binary or categorical. Only age and family income are non-categorical features in the raw datasets. In order to binarize these features we threshold on the mean age (family income) of the dataset and define categories of Young (Low Income) and Old (High Income).

All analyses presented in this work were performed on an Apple M1 Max processor with 64 GB memory. Source code was executed in Python 3.10.12, with notable libraries including NumPy 1.26.4, SciPy 1.13.0, scikit-learn 1.4.2, and Pandas 2.2.2. Full source code and results for this project may be accessed at `https://anonymous.4open.science/r/rep_fairness-5A05`

## C  Full Experimental Results

### C.1  Arm-Based Sampling

In figure 5(a,d,g,j), we show the representativeness of the dataset constructed over time by each approach in a no-bias situation. While performance is similar between contemporary baseline algorithms, D-PBRS yields the most representative samples, often approaching the fully informed algorithm OPT. Significant response bias in either direction harms the representativeness of the final cohort (Fig. 5(b,e,h,k)). Yet, D-PBRS, and to a lesser extent PBRS, consistently yield more representative datasets than other under cases of response bias. Figure 5(c,f,i,l) depicts dataset representativeness as a function of the casual bias ($\alpha$); as $\alpha$ increases, sampling a site increases the probability the member of majority groups will appear in future samples from that site. Similar to response bias, representativeness decreases as the bias becomes more pronounced. Unlike response bias, causal bias results in distribution shifts over time, increasing the difficult of accurately assessing which arm is best to sample. Due to this shift, the advantage of PBRS and D-PBRS over other algorithms diminishes, though D-PBRS still remains the best performing.

### C.2  Direct Sampling and Fairness

We present analyses for the population and group-wise AUC, TPR, and TNR of classifiers trained on datasets which vary in proportions of each sensitive feature. Figures 6-11 show population, and group-wise, performance as a function the fraction of samples in the training data which are from $G_0$ and $G_1$ (shown on each plot). The methodology and presentation of these results parallels main body section 5 figure 1. There is significant TPR and TNR unfairness in the Law School, Texas Salary, Adult Income, and Community Crime data domains.

### C.3  Effects of Realistic Sampling Techniques

Extending the results from section 6, we present results for arm-based sampling methods (PBRS, D-PBRS, OPT, D-OPT) alongside SRS for all data domains with location variables (Figs. 12-31). As discussed in the main body, there are two key observations in these figures. First, an increase in the representation of a given group may, but does not always, significantly improve downstream performance on that group even in the SRS case. Second, the sampling method plays a crucial role in the relationship between downstream fairness and representation. The arm-based sampling methods often show very different subgroup performance than SRS, though they tend to be similar to each other.

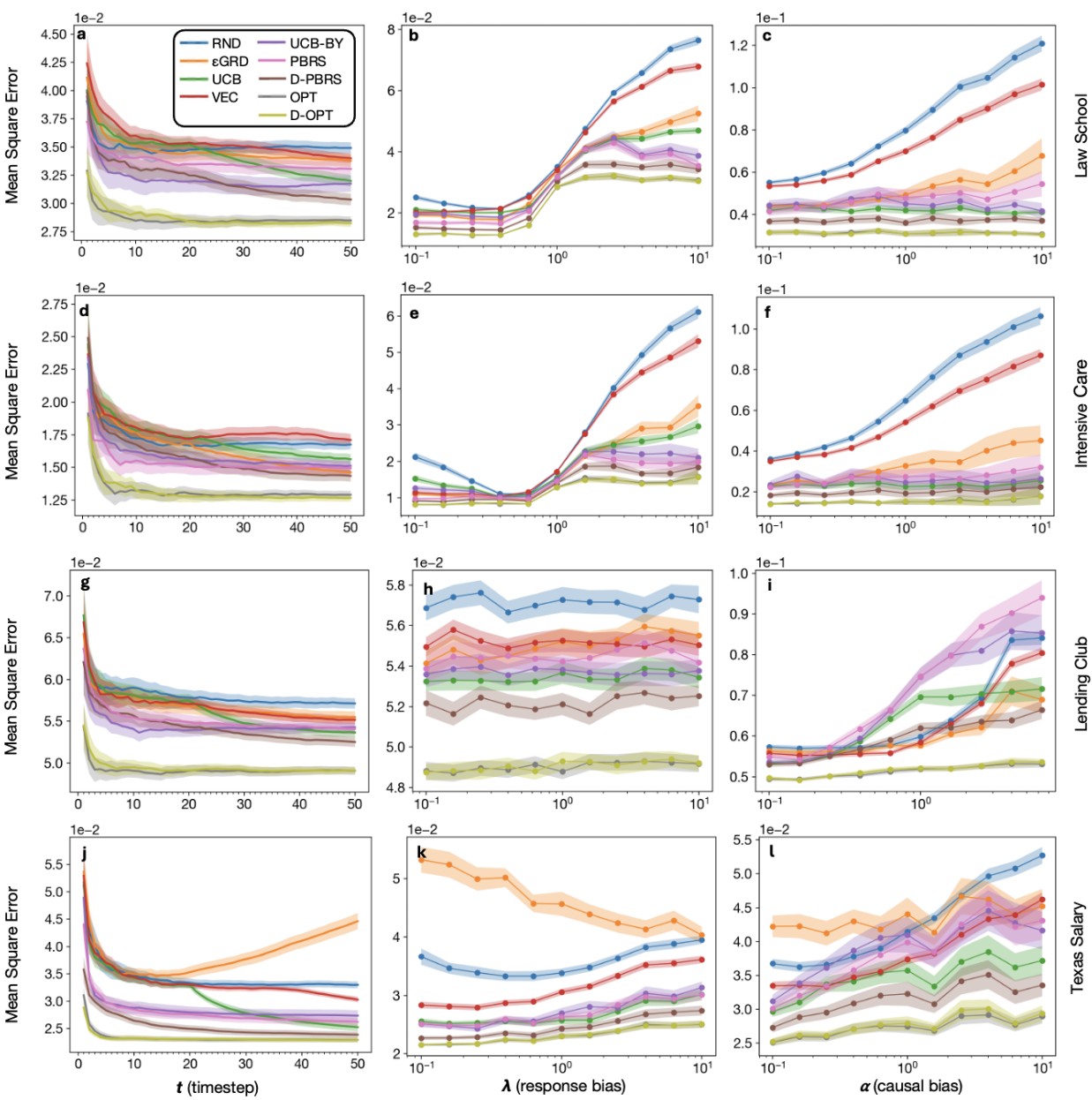

Figure 5: Dataset representativeness in all datasets measured by distance between cohort sensitive feature means and target vector $\boldsymbol{v} = \langle .5, \cdots, .5 \rangle$ as the cohort is constructed the no-bias case **(a), (d), (g), (j)**, for the final cohort in the non-causal response bias case **(b), (e), (h), (k)**, and for the final cohort in the causal distribution shift case **(c), (f), (i), (l)**. Two of our proposed algorithms (UCB-BY and PBRS) perform comparably to state-of-the-art sampling algorithms while D-PBRS outperforms all sampling algorithms except the fully-informed gold-standard OPT. Shaded regions indicate 95% confidence intervals.

## C.4 Complexity Analysis

We include expanded results for complexity analyses of all sensitive features on selected datasets known to have unfairness (Law School, Adult Income, Community Crime). In these figures, we plot each experimental combination of complexity and representation as a scatterplot point with $\Delta$TPR and $\Delta$TNR on the x- and y-axis, respectively (Figs. 32-39). Parity of TPR and TNR (i.e., $\Delta$TPR = $\Delta$TNR = 0) represents optimal

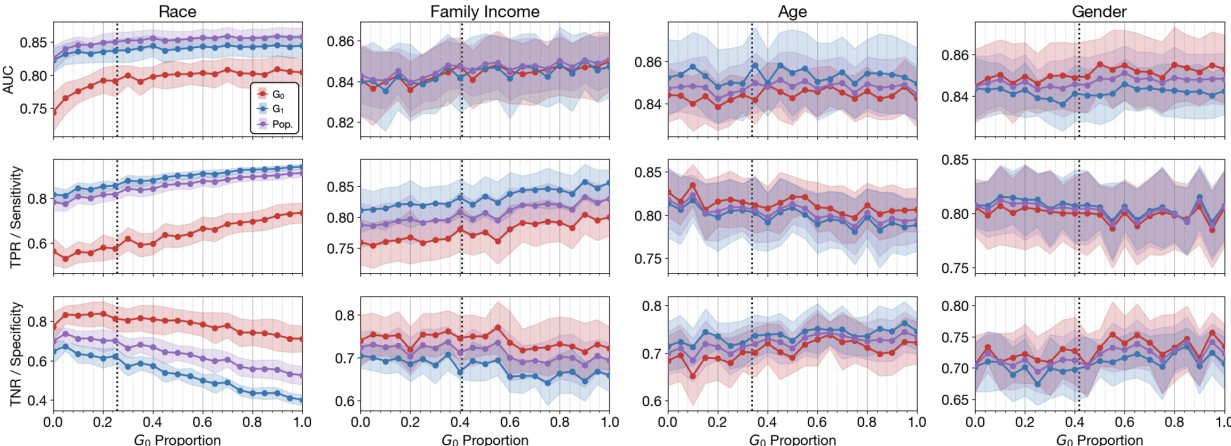

Figure 6: Population (purple) and subgroup (red and blue) AUCs, TPRs, and TNRs for gradient-boosted classifiers in the Law School data domain. Each column represents an analysis studying group proportions by one sensitive feature. Datasets are selected by SRS from the entire data domain, with no location restrictions.

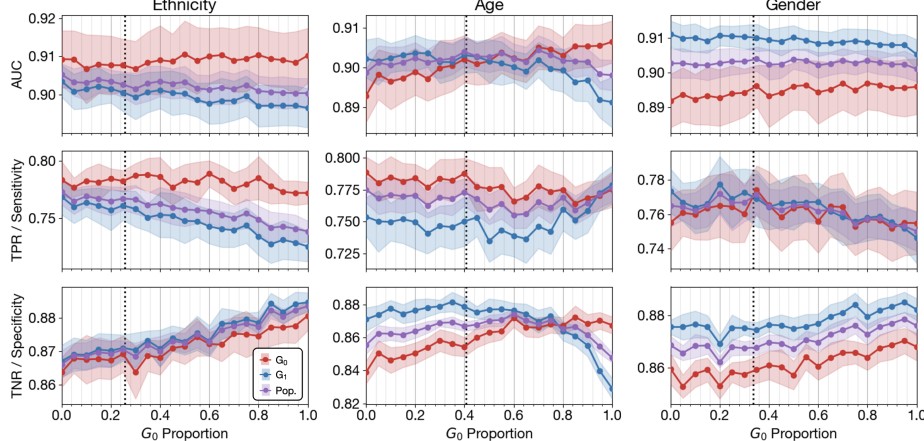

Figure 7: Population (purple) and subgroup (red and blue) AUCs, TPRs, and TNRs for gradient-boosted classifiers in the Intensive Care data domain. Each column represents an analysis studying group proportions by one sensitive feature. Datasets are selected by SRS from the entire data domain, with no location restrictions.

fairness and is indicated by the intersection of the two dotted black lines. We repeat the same scatterplot three times, color-coding the points differentially to highlight different experimental facets. The leftmost panel shows training set proportions, the middle panel shows model complexity, and the rightmost panel shows population AUC. Complexity is assessed through a combination of maximum tree depth and number of estimation steps: as both hyperparameters increase, do does complexity. The plotted combinations of complexity, in form of (max_depth), (n_estimators) tuples ordered from least complexity to greatest, are:

1. (1,1)

2. (1,5)

3. (1,10)

4. (1,20)

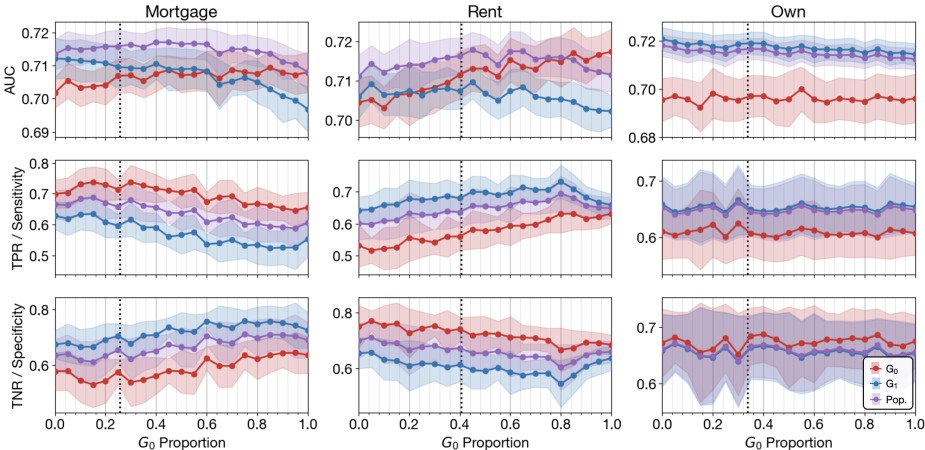

Figure 8: Population (purple) and subgroup (red and blue) AUCs, TPRs, and TNRs for gradient-boosted classifiers in the Lending Club data domain. Each column represents an analysis studying group proportions by one sensitive feature. Datasets are selected by SRS from the entire data domain, with no location restrictions.

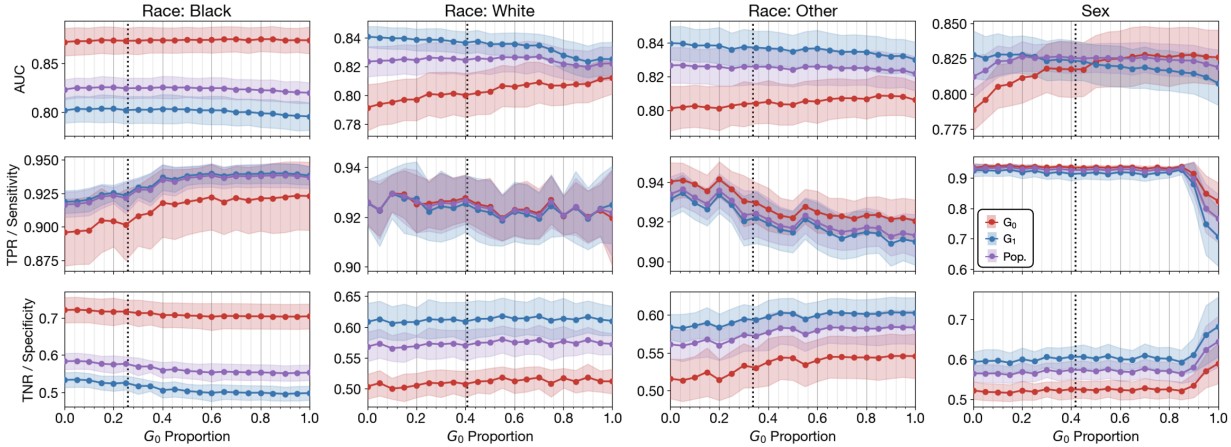

Figure 9: Population (purple) and subgroup (red and blue) AUCs, TPRs, and TNRs for gradient-boosted classifiers in the Texas Salary data domain. Each column represents an analysis studying group proportions by one sensitive feature. Datasets are selected by SRS from the entire data domain, with no location restrictions.

5. (2,30)

6. (2,40)

7. (2,50)

8. (3,60)

9. (3,80)

10. (3,90)

11. **(3,100)**

12. (4,100)

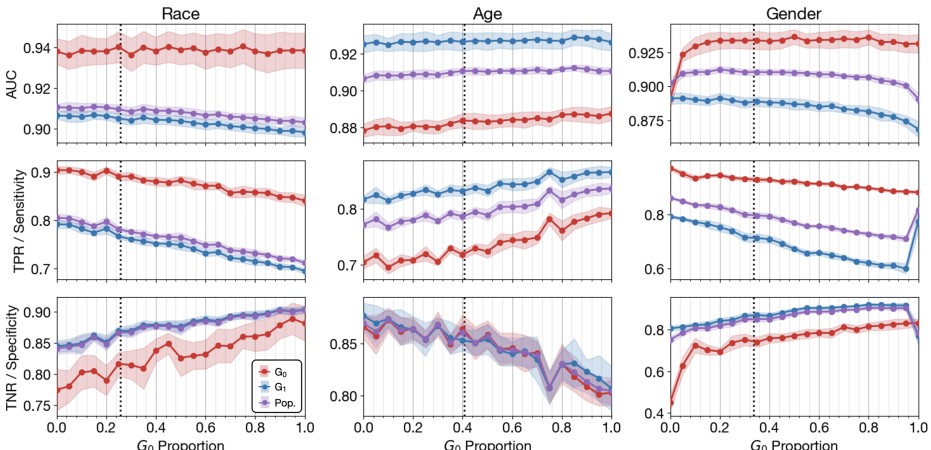

Figure 10: Population (purple) and subgroup (red and blue) AUCs, TPRs, and TNRs for gradient-boosted classifiers in the Adult Income data domain. Each column represents an analysis studying group proportions by one sensitive feature. Datasets are selected by SRS from the entire data domain, with no location restrictions.

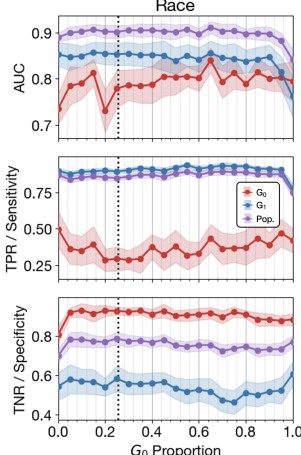

Figure 11: Population (purple) and subgroup (red and blue) AUCs, TPRs, and TNRs for gradient-boosted classifiers in the Community Crime data domain. Each column represents an analysis studying group proportions by one sensitive feature. Datasets are selected by SRS from the entire data domain, with no location restrictions.

13. (4,200)

14. (5,200)

15. (5,300)

16. (6,300)

17. (7,300)

18. (7,400)

19. (8,400)

20. (8,500)

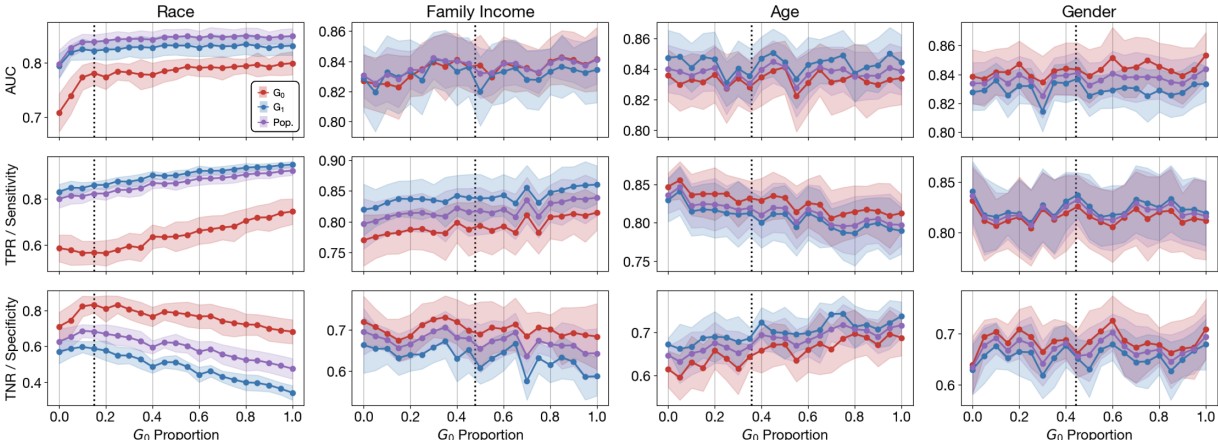

Figure 12: Population (purple) and subgroup (red and blue) AUCs, TPRs, and TNRs for GBCs in the Law School data domain. Each column represents an analysis studying group proportions by one sensitive feature. Datasets are selected by SRS from the location-filtered data domain.

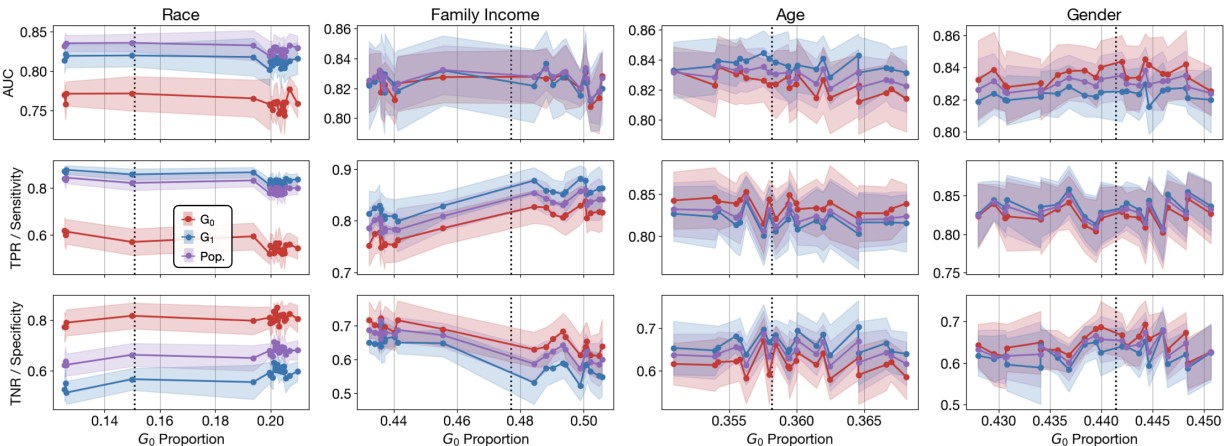

Figure 13: Population (purple) and subgroup (red and blue) AUCs, TPRs, and TNRs for GBCs in the Law School data domain. Each column represents an analysis studying group proportions by one sensitive feature. Datasets are selected by PBRS from the location-filtered data domain.

We bold the combination (3,100) as this is the default hyperparameter setting.

In general, increased model complexity improves fairness through better AUC, TPR, and/or TNR parity (primarily via TPR parity). While there are a couple cases where increasing model complexity does not significantly improve fairness, it does not appear that increasing model complexity *harms* fairness, making it at least a potentially beneficial intervention from a fairness perspective. As discussed in the main body, improvements in algorithmic fairness can often come at the cost of overall classifier performance. To analyze whether any fairness gains we see from increased model complexity harm classifier performance, we show the overall test set AUC of models with varying complexity. In general, overall classifier AUC peaks at intermediate model complexity. Higher complexity levels sometimes show moderate degradation in performance, but substantial fairness gains can be realized at lower complexity levels with no accuracy penalty.

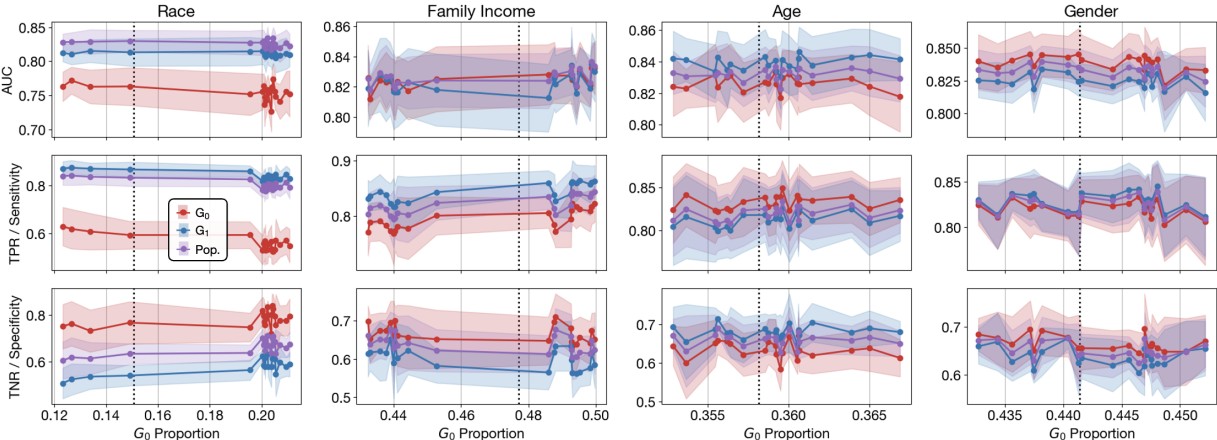

Figure 14: Population (purple) and subgroup (red and blue) AUCs, TPRs, and TNRs for GBCs in the Law School data domain. Each column represents an analysis studying group proportions by one sensitive feature. Datasets are selected by OPT from the location-filtered data domain.

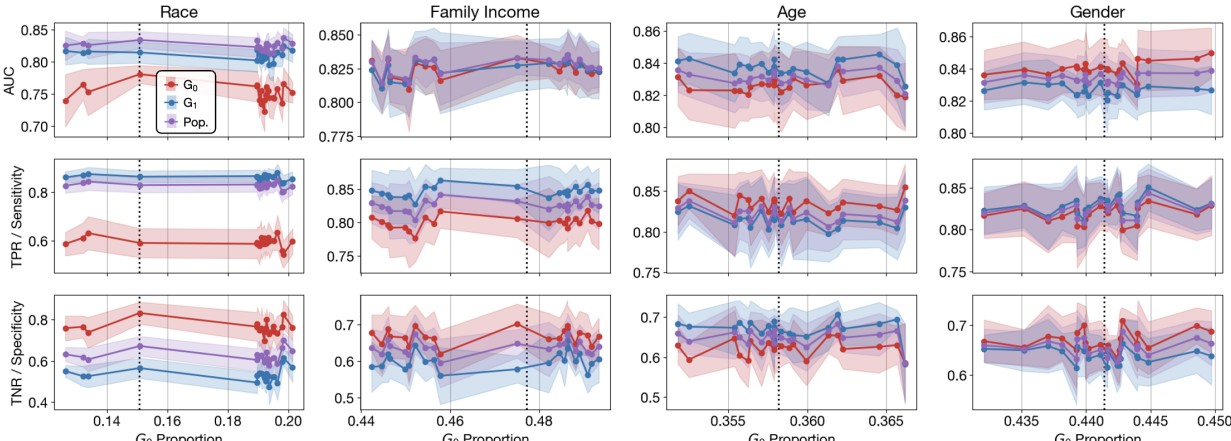

Figure 15: Population (purple) and subgroup (red and blue) AUCs, TPRs, and TNRs for GBCs in the Law School data domain. Each column represents an analysis studying group proportions by one sensitive feature. Datasets are selected by D-PBRS from the location-filtered data domain.

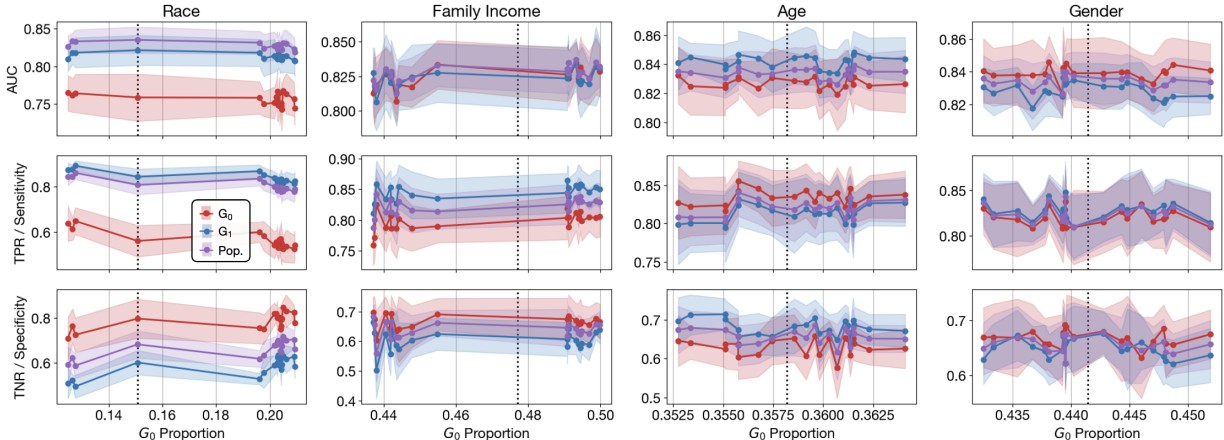

Figure 16: Population (purple) and subgroup (red and blue) AUCs, TPRs, and TNRs for GBCs in the Law School data domain. Each column represents an analysis studying group proportions by one sensitive feature. Datasets are selected by D-OPT from the location-filtered data domain.

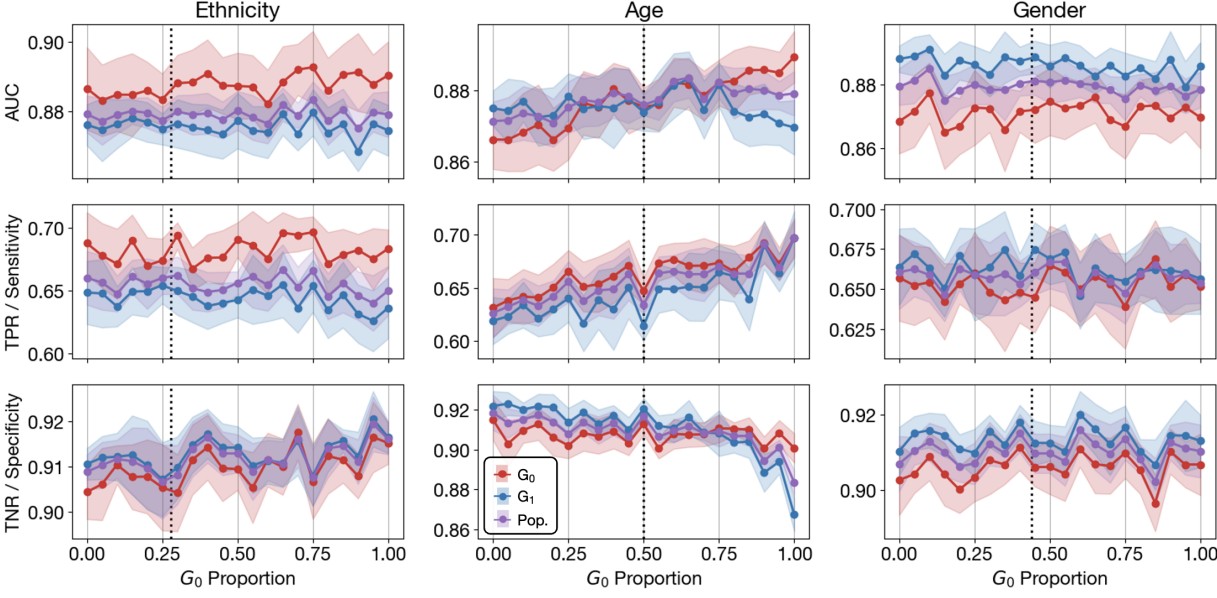

Figure 17: Population (purple) and subgroup (red and blue) AUCs, TPRs, and TNRs for GBCs in the Intensive Care data domain. Each column represents an analysis studying group proportions by one sensitive feature. Datasets are selected by SRS from the location-filtered data domain.

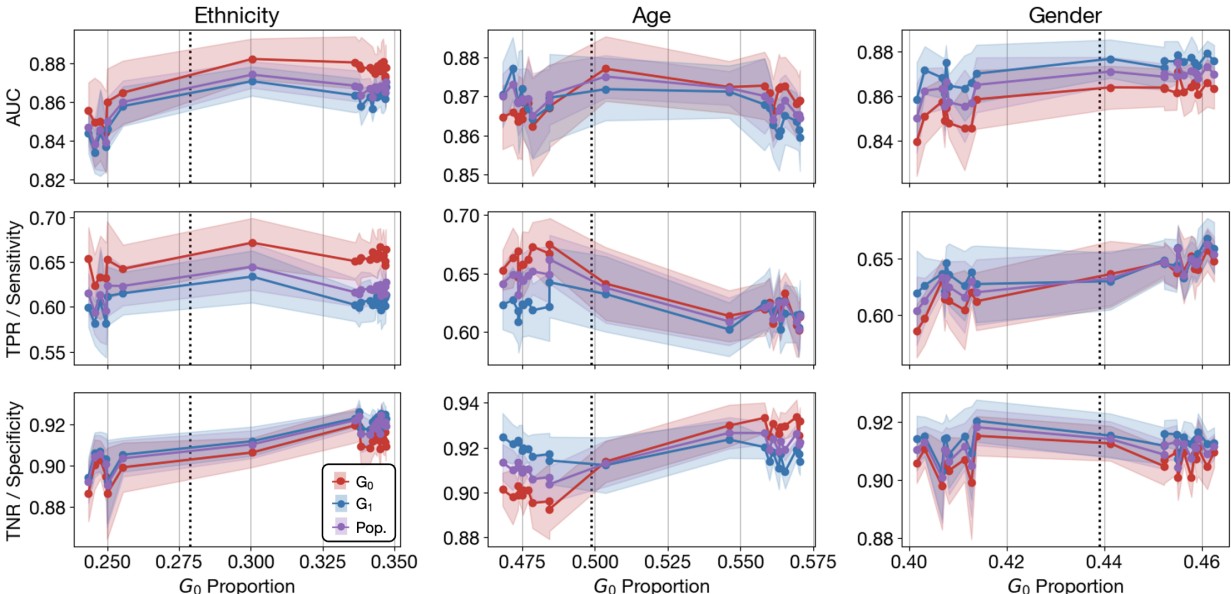

Figure 18: Population (purple) and subgroup (red and blue) AUCs, TPRs, and TNRs for GBCs in the Intensive Care data domain. Each column represents an analysis studying group proportions by one sensitive feature. Datasets are selected by PBRS from the location-filtered data domain.

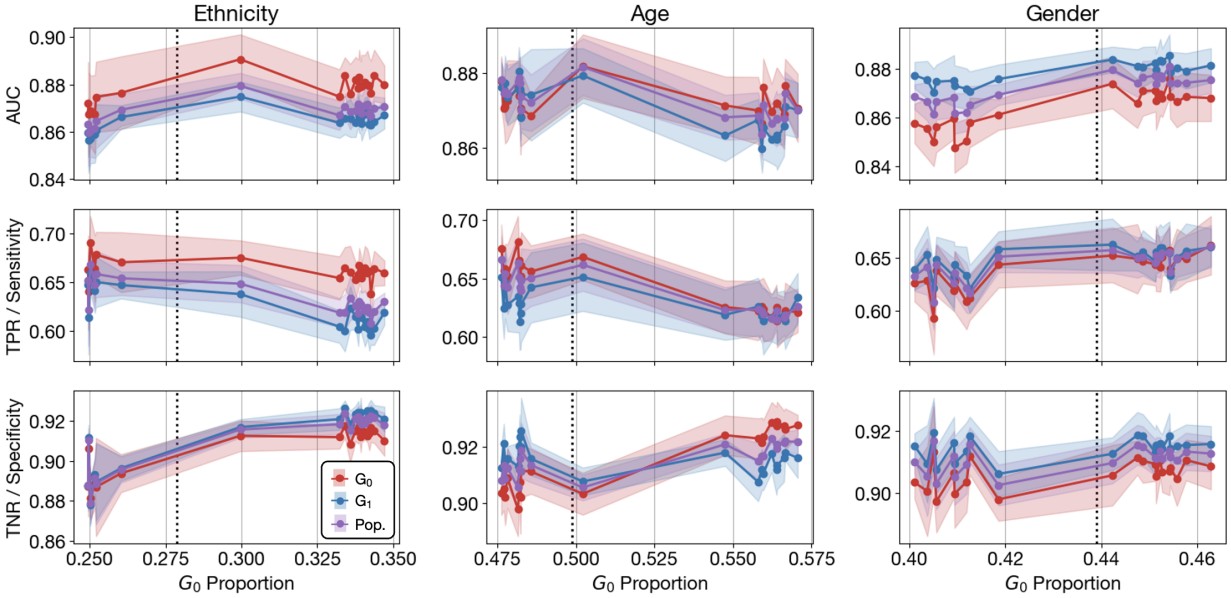

Figure 19: Population (purple) and subgroup (red and blue) AUCs, TPRs, and TNRs for GBCs in the Intensive Care data domain. Each column represents an analysis studying group proportions by one sensitive feature. Datasets are selected by OPT from the location-filtered data domain.

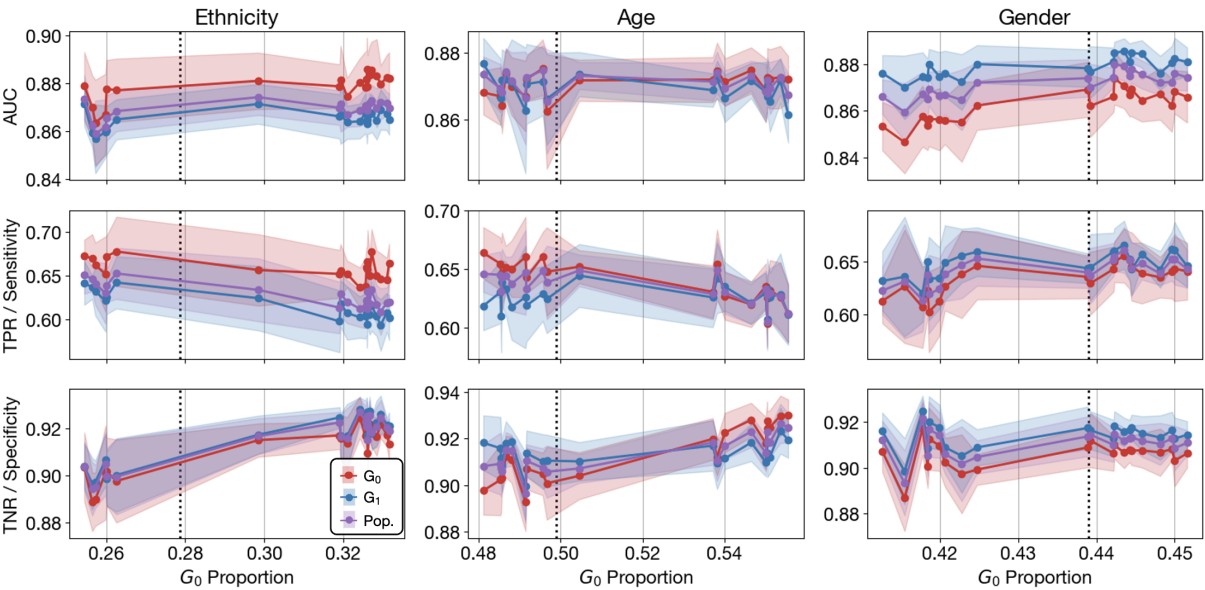

Figure 20: Population (purple) and subgroup (red and blue) AUCs, TPRs, and TNRs for GBCs in the Intensive Care data domain. Each column represents an analysis studying group proportions by one sensitive feature. Datasets are selected by D-PBRS from the location-filtered data domain.

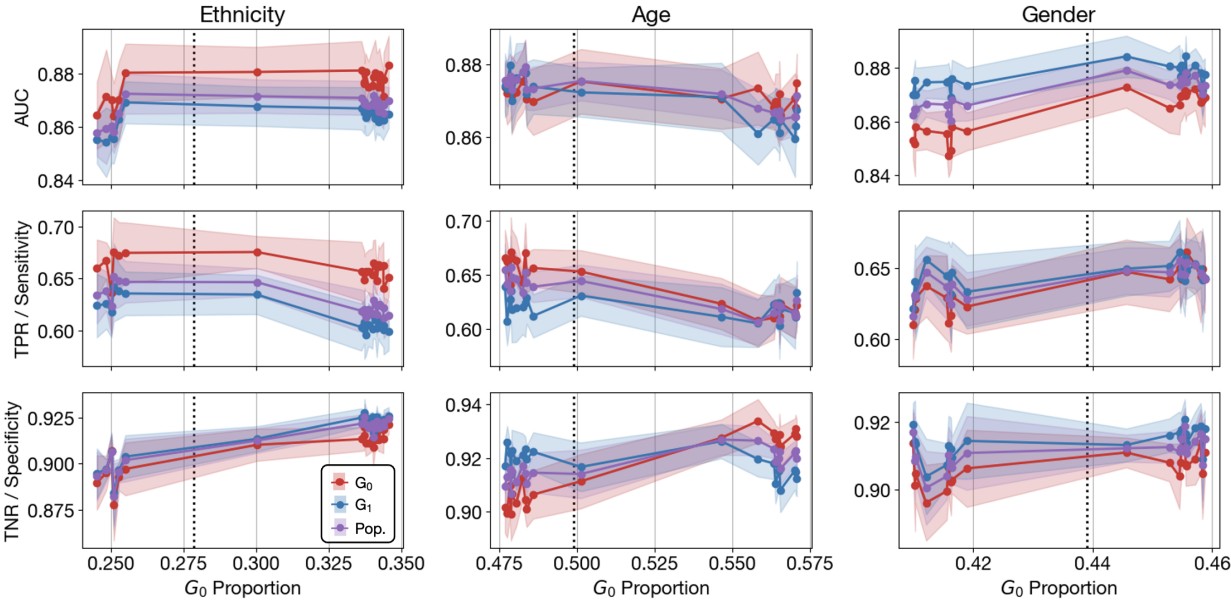

Figure 21: Population (purple) and subgroup (red and blue) AUCs, TPRs, and TNRs for GBCs in the Intensive Care data domain. Each column represents an analysis studying group proportions by one sensitive feature. Datasets are selected by D-OPT from the location-filtered data domain.

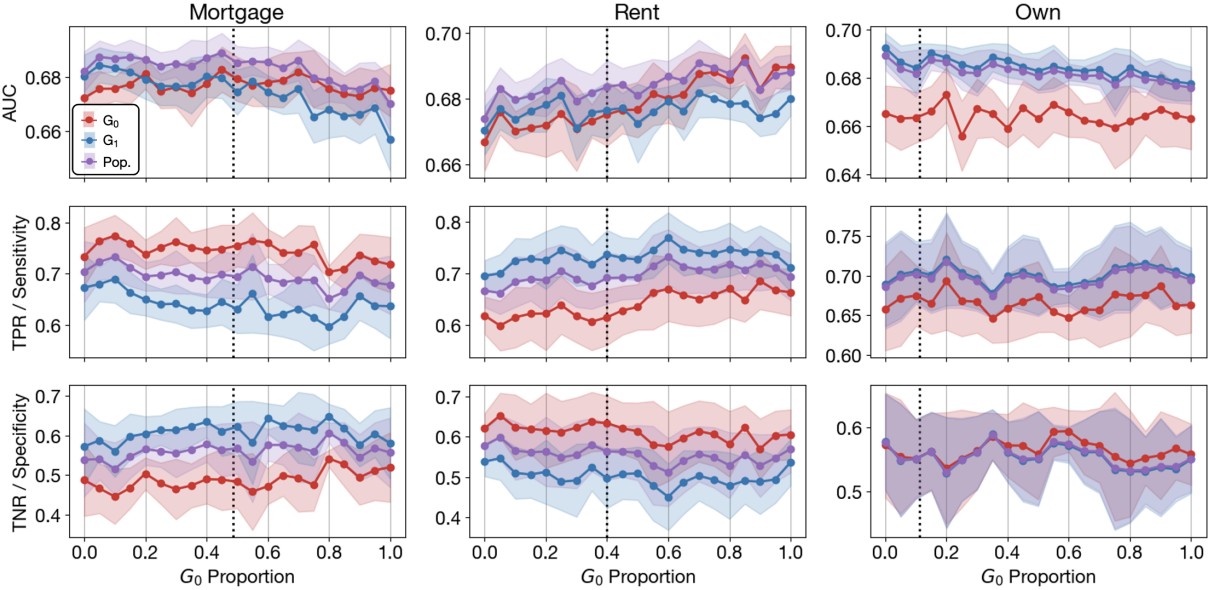

Figure 22: Population (purple) and subgroup (red and blue) AUCs, TPRs, and TNRs for GBCs in the Lending Club data domain. Each column represents an analysis studying group proportions by one sensitive feature. Datasets are selected by SRS from the location-filtered data domain.

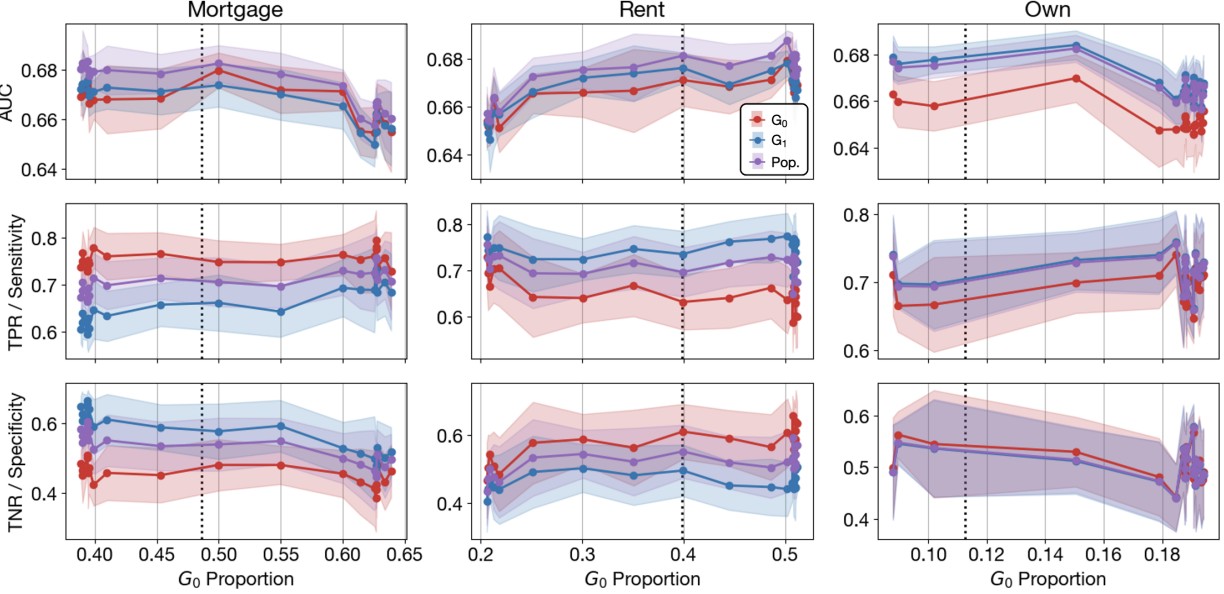

Figure 23: Population (purple) and subgroup (red and blue) AUCs, TPRs, and TNRs for GBCs in the Lending Club data domain. Each column represents an analysis studying group proportions by one sensitive feature. Datasets are selected by PBRS from the location-filtered data domain.

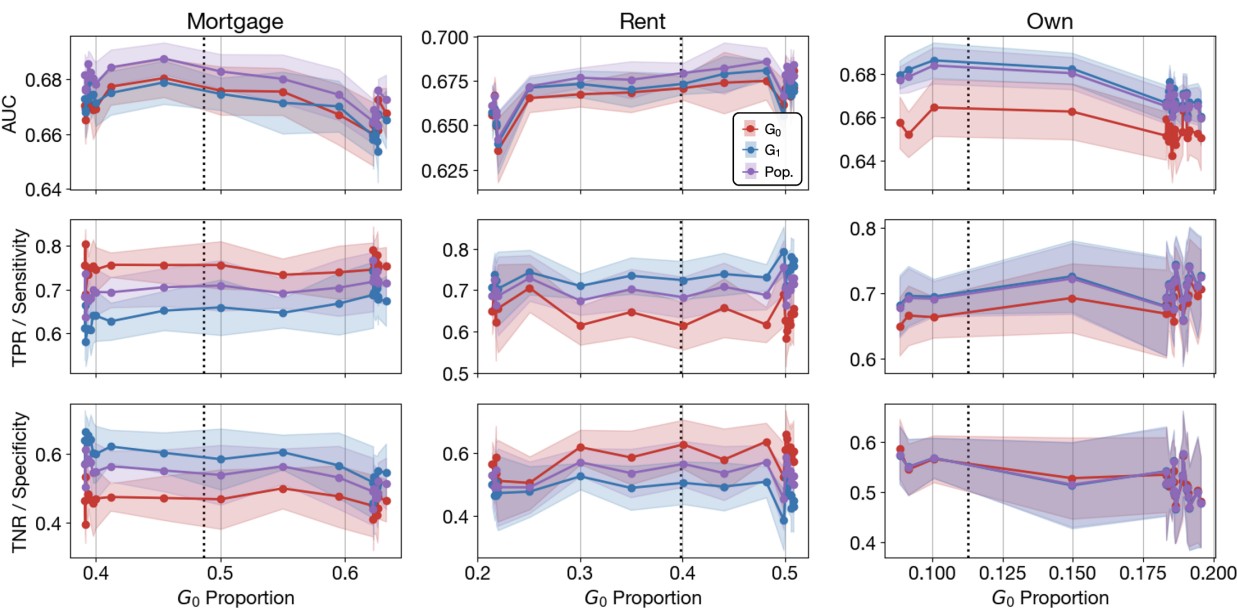

Figure 24: Population (purple) and subgroup (red and blue) AUCs, TPRs, and TNRs for GBCs in the Lending Club data domain. Each column represents an analysis studying group proportions by one sensitive feature. Datasets are selected by OPT from the location-filtered data domain.

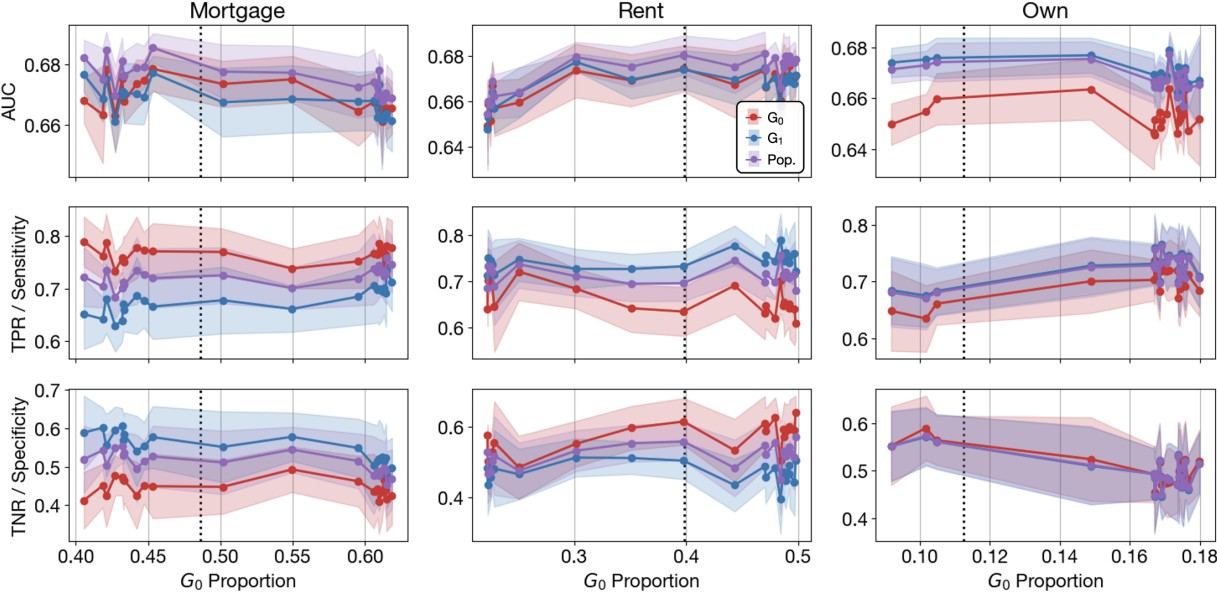

Figure 25: Population (purple) and subgroup (red and blue) AUCs, TPRs, and TNRs for GBCs in the Lending Club data domain. Each column represents an analysis studying group proportions by one sensitive feature. Datasets are selected by D-PBRS from the location-filtered data domain.

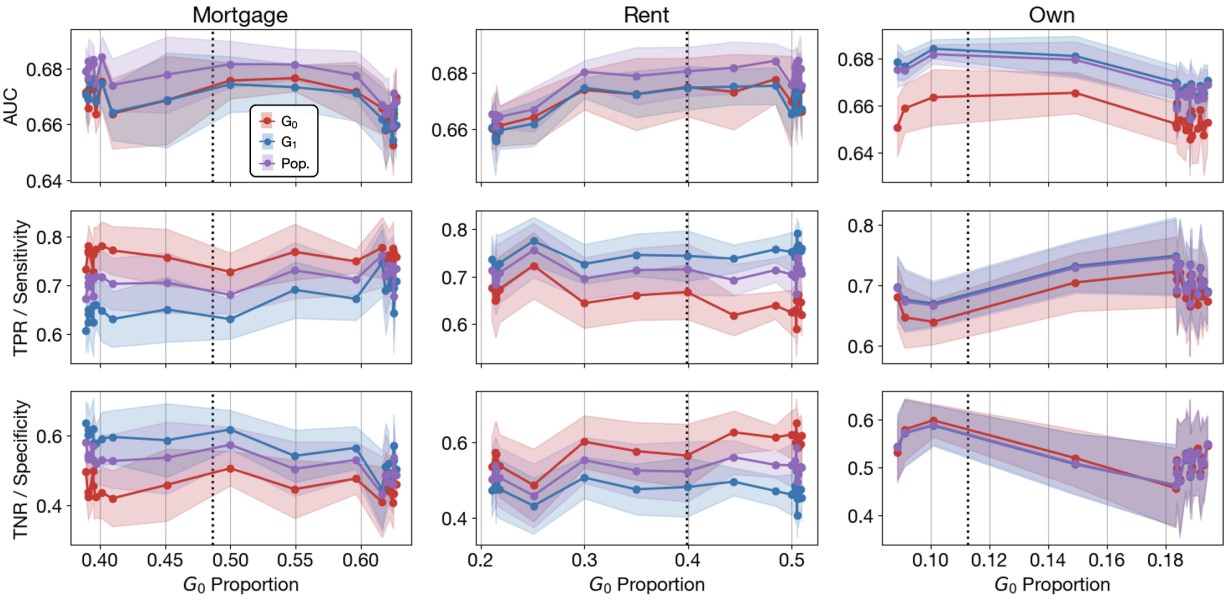

Figure 26: Population (purple) and subgroup (red and blue) AUCs, TPRs, and TNRs for GBCs in the Lending Club data domain. Each column represents an analysis studying group proportions by one sensitive feature. Datasets are selected by D-OPT from the location-filtered data domain.

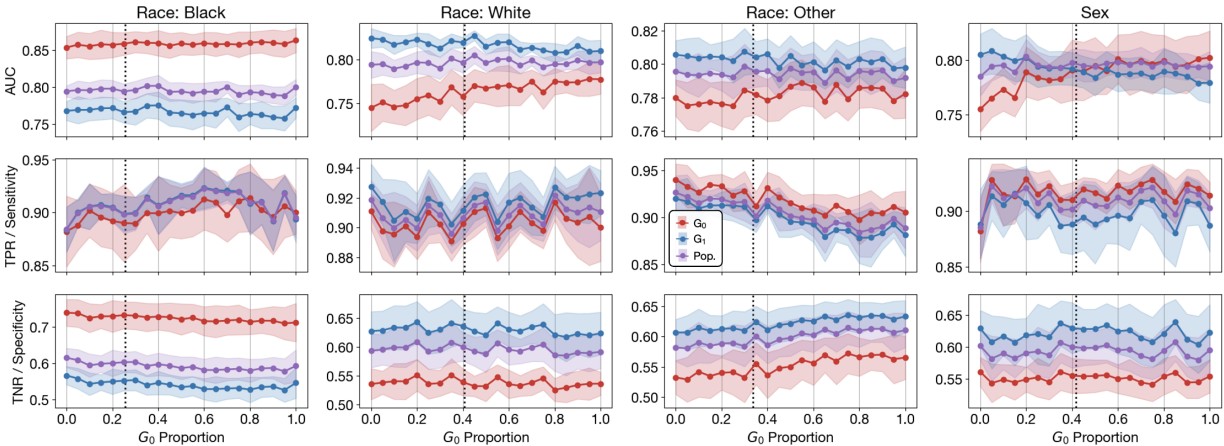

Figure 27: Population (purple) and subgroup (red and blue) AUCs, TPRs, and TNRs for GBCs in the Texas Salary data domain. Each column represents an analysis studying group proportions by one sensitive feature. Datasets are selected by SRS from the location-filtered data domain.

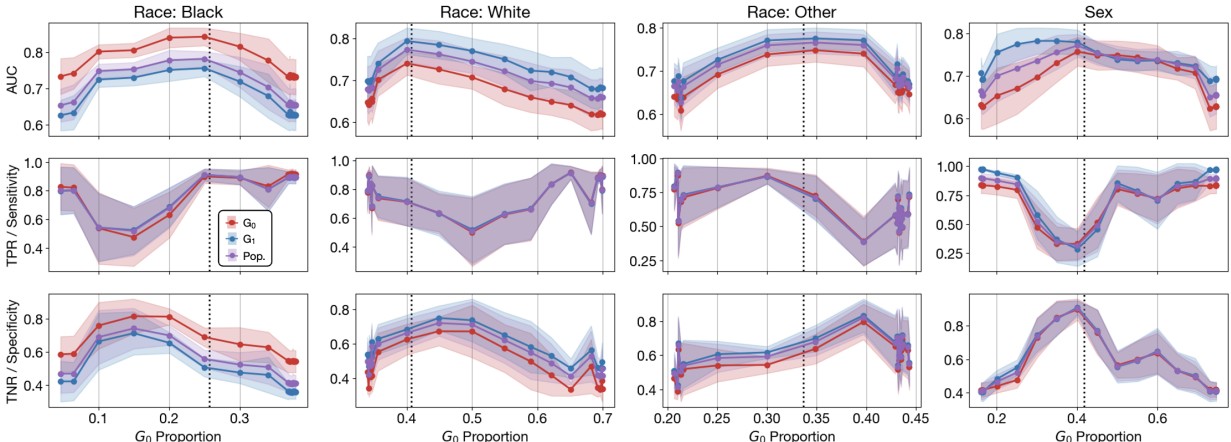

Figure 28: Population (purple) and subgroup (red and blue) AUCs, TPRs, and TNRs for GBCs in the Texas Salary data domain. Each column represents an analysis studying group proportions by one sensitive feature. Datasets are selected by PBRS from the location-filtered data domain.

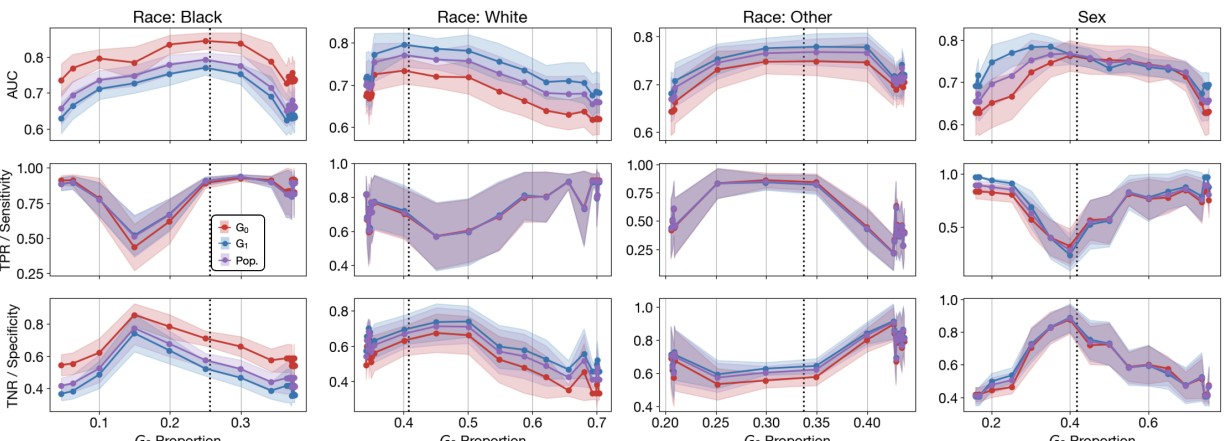

Figure 29: Population (purple) and subgroup (red and blue) AUCs, TPRs, and TNRs for GBCs in the Texas Salary data domain. Each column represents an analysis studying group proportions by one sensitive feature. Datasets are selected by OPT from the location-filtered data domain.

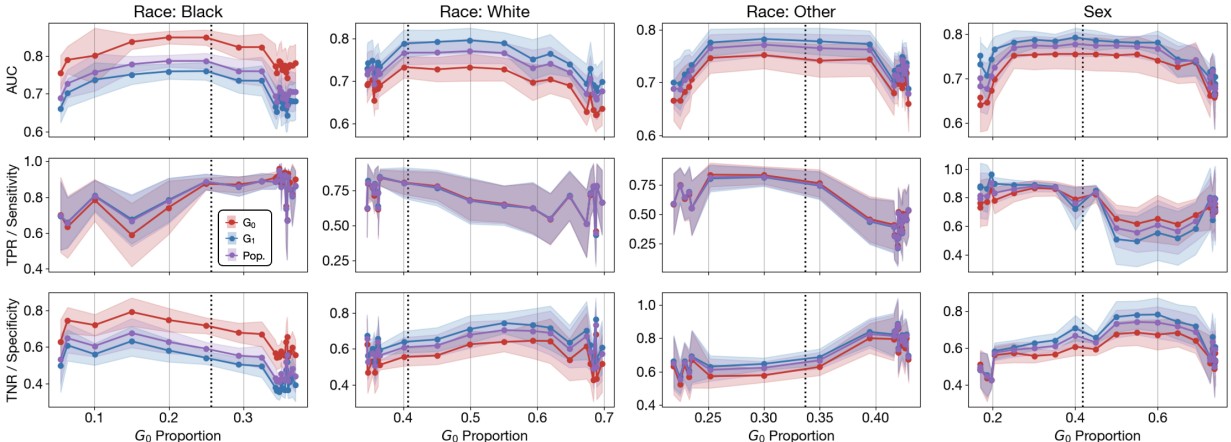

Figure 30: Population (purple) and subgroup (red and blue) AUCs, TPRs, and TNRs for GBCs in the Texas Salary data domain. Each column represents an analysis studying group proportions by one sensitive feature. Datasets are selected by D-PBRS from the location-filtered data domain.

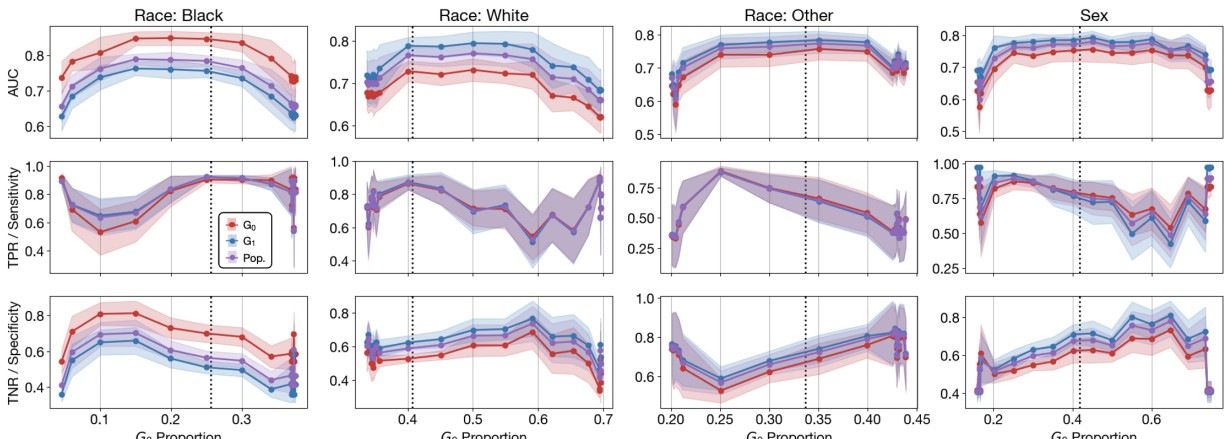

Figure 31: Population (purple) and subgroup (red and blue) AUCs, TPRs, and TNRs for GBCs in the Texas Salary data domain. Each column represents an analysis studying group proportions by one sensitive feature. Datasets are selected by D-OPT from the location-filtered data domain.

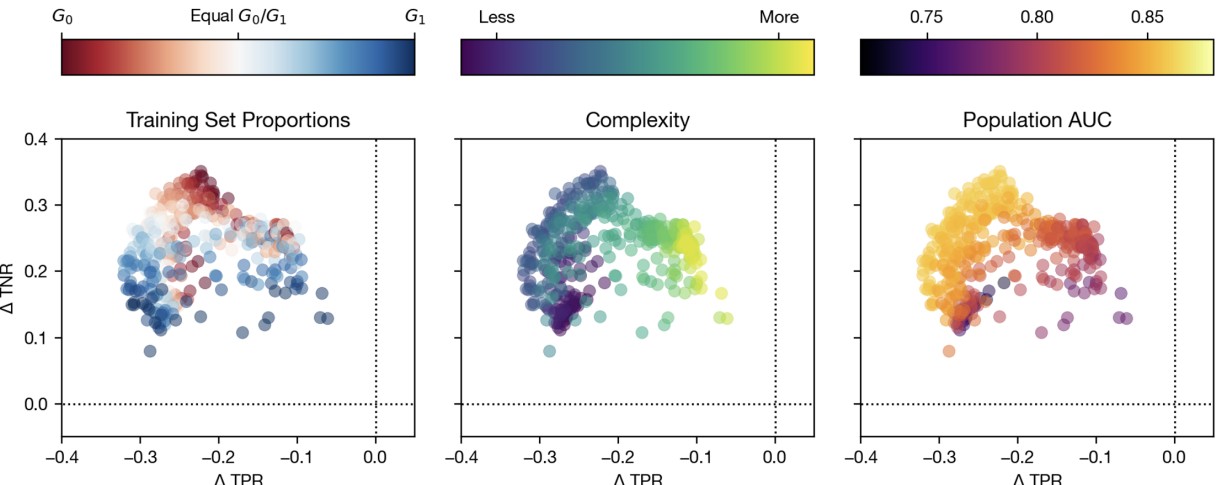

Figure 32: TPR and TNR unfairness at varying levels of group representativeness and complexity for the Law School data domain analyzed by race. Optimal fairness is shown by the intersection of the two dotted black lines. The leftmost panel is color-coded by group representation (deepest red indicating primarily $G_0$, deepest blue indicating primarily $G_1$, and white indicating balance. The middle panel is color-coded by model complexity, with lighter values indicating higher model complexity. The rightmost panel is color-coded by population AUC, with lighter values indicating higher performance.

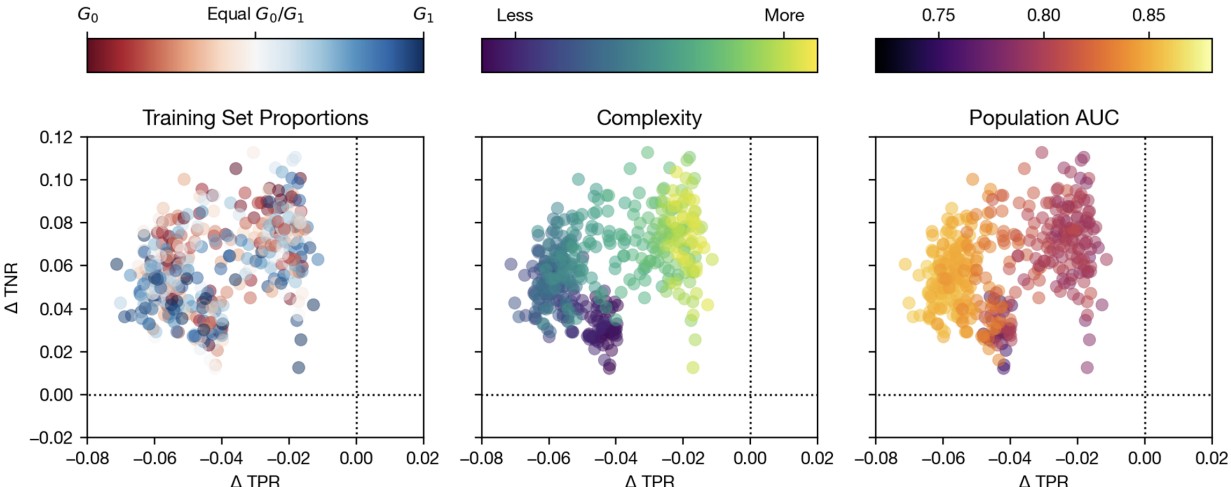

Figure 33: TPR and TNR unfairness at varying levels of group representativeness and complexity for the Law School data domain analyzed by family income. Optimal fairness is shown by the intersection of the two dotted black lines. The leftmost panel is color-coded by group representation (deepest red indicating primarily $G_0$, deepest blue indicating primarily $G_1$, and white indicating balance. The middle panel is color-coded by model complexity, with lighter values indicating higher model complexity. The rightmost panel is color-coded by population AUC, with lighter values indicating higher performance.

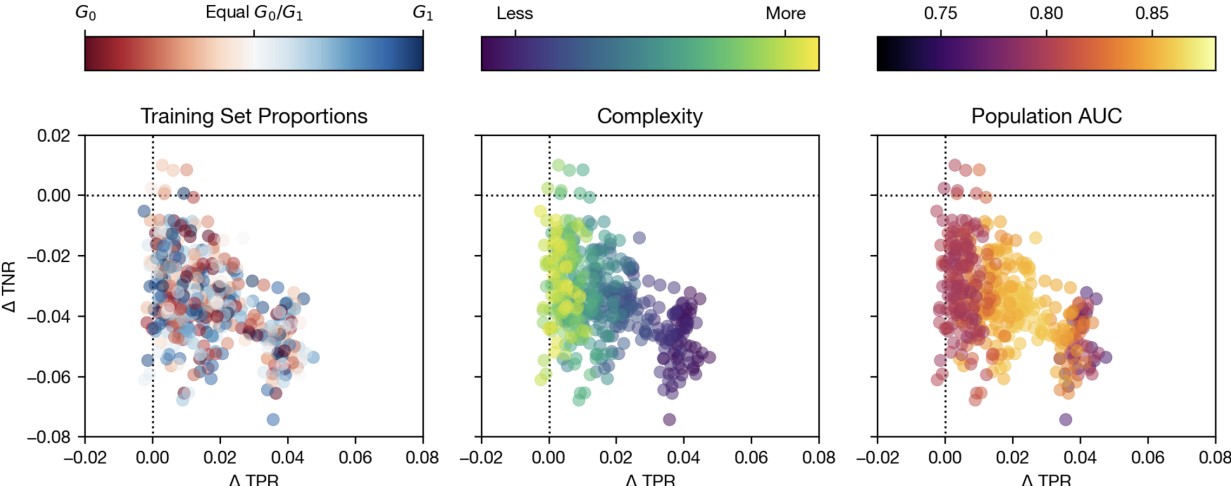

Figure 34: TPR and TNR unfairness at varying levels of group representativeness and complexity for the Law School data domain analyzed by age. Optimal fairness is shown by the intersection of the two dotted black lines. The leftmost panel is color-coded by group representation (deepest red indicating primarily $G_0$, deepest blue indicating primarily $G_1$, and white indicating balance. The middle panel is color-coded by model complexity, with lighter values indicating higher model complexity. The rightmost panel is color-coded by population AUC, with lighter values indicating higher performance.

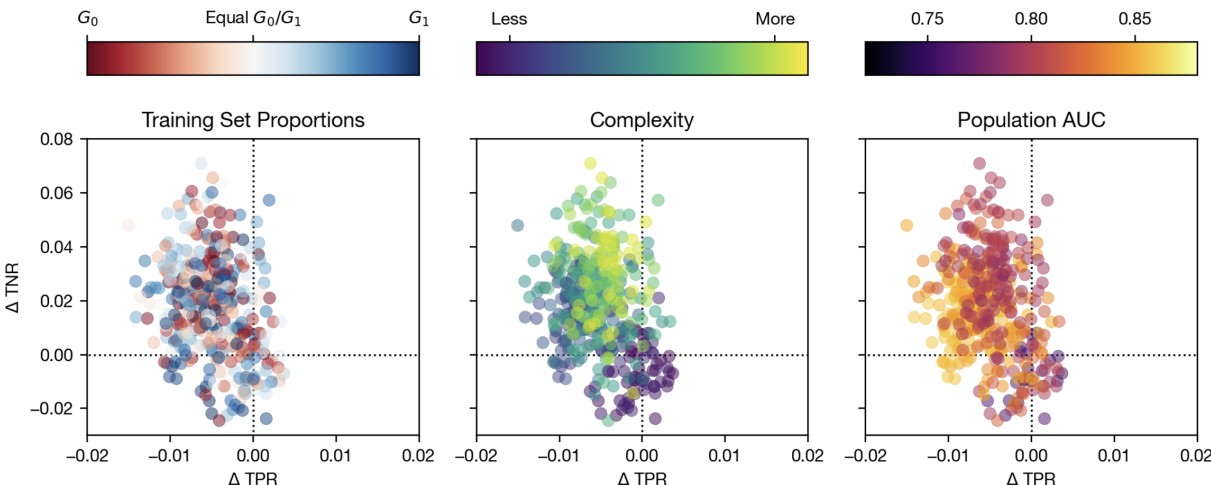

Figure 35: TPR and TNR unfairness at varying levels of group representativeness and complexity for the Law School data domain analyzed by gender. Optimal fairness is shown by the intersection of the two dotted black lines. The leftmost panel is color-coded by group representation (deepest red indicating primarily $G_0$, deepest blue indicating primarily $G_1$, and white indicating balance. The middle panel is color-coded by model complexity, with lighter values indicating higher model complexity. The rightmost panel is color-coded by population AUC, with lighter values indicating higher performance.

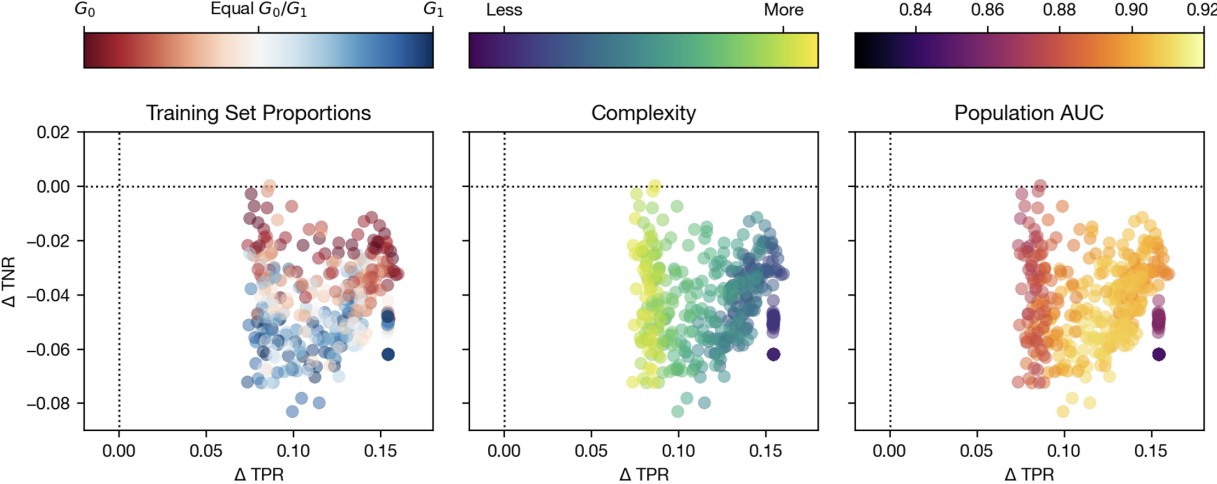

Figure 36: TPR and TNR unfairness at varying levels of group representativeness and complexity for the Adult Income data domain analyzed by race. Optimal fairness is shown by the intersection of the two dotted black lines. The leftmost panel is color-coded by group representation (deepest red indicating primarily $G_0$, deepest blue indicating primarily $G_1$, and white indicating balance. The middle panel is color-coded by model complexity, with lighter values indicating higher model complexity. The rightmost panel is color-coded by population AUC, with lighter values indicating higher performance.

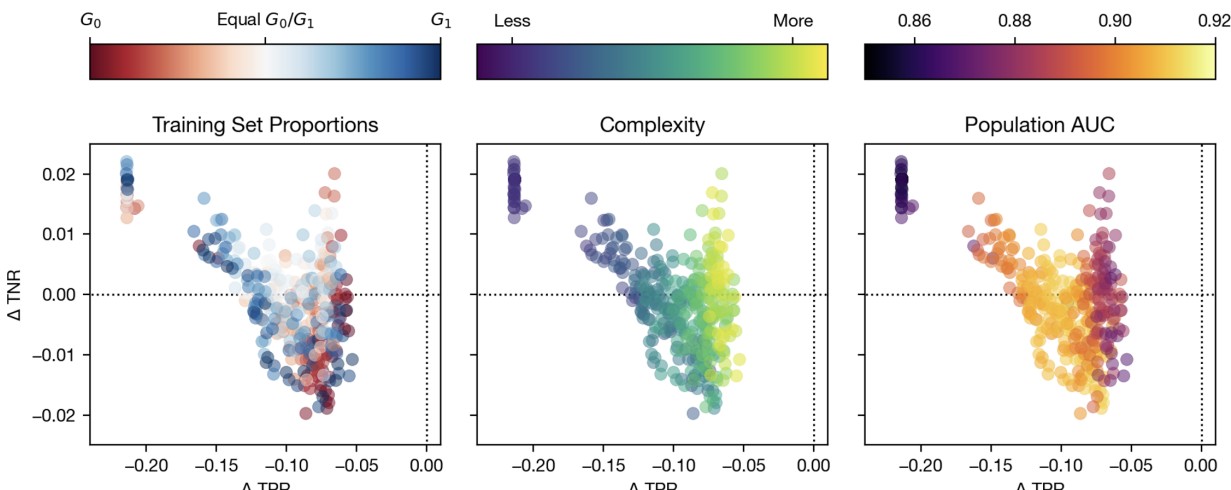

Figure 37: TPR and TNR unfairness at varying levels of group representativeness and complexity for the Adult Income data domain analyzed by age. Optimal fairness is shown by the intersection of the two dotted black lines. The leftmost panel is color-coded by group representation (deepest red indicating primarily $G_0$, deepest blue indicating primarily $G_1$, and white indicating balance. The middle panel is color-coded by model complexity, with lighter values indicating higher model complexity. The rightmost panel is color-coded by population AUC, with lighter values indicating higher performance.

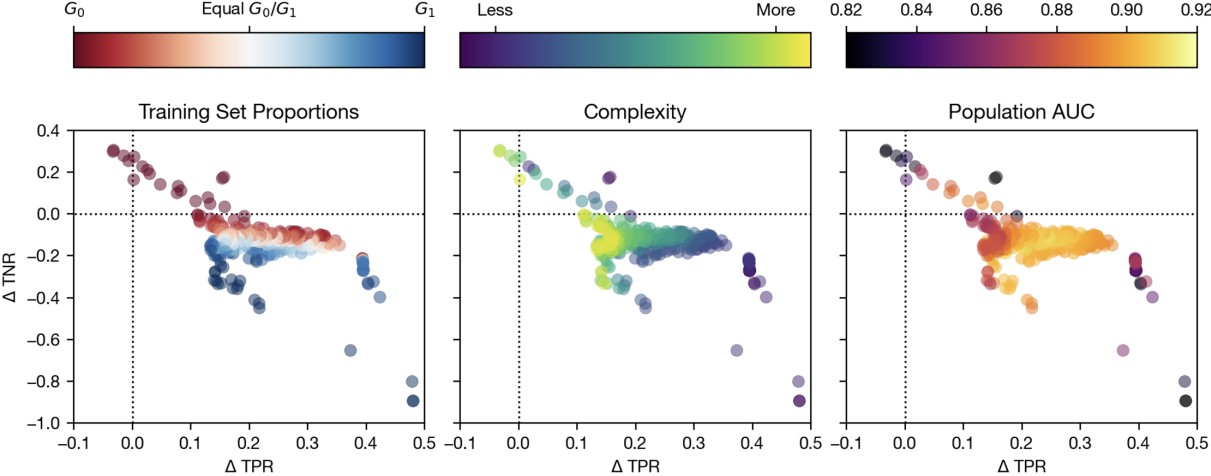

Figure 38: TPR and TNR unfairness at varying levels of group representativeness and complexity for the Adult Income data domain analyzed by gender. Optimal fairness is shown by the intersection of the two dotted black lines. The leftmost panel is color-coded by group representation (deepest red indicating primarily $G_0$, deepest blue indicating primarily $G_1$, and white indicating balance. The middle panel is color-coded by model complexity, with lighter values indicating higher model complexity. The rightmost panel is color-coded by population AUC, with lighter values indicating higher performance.

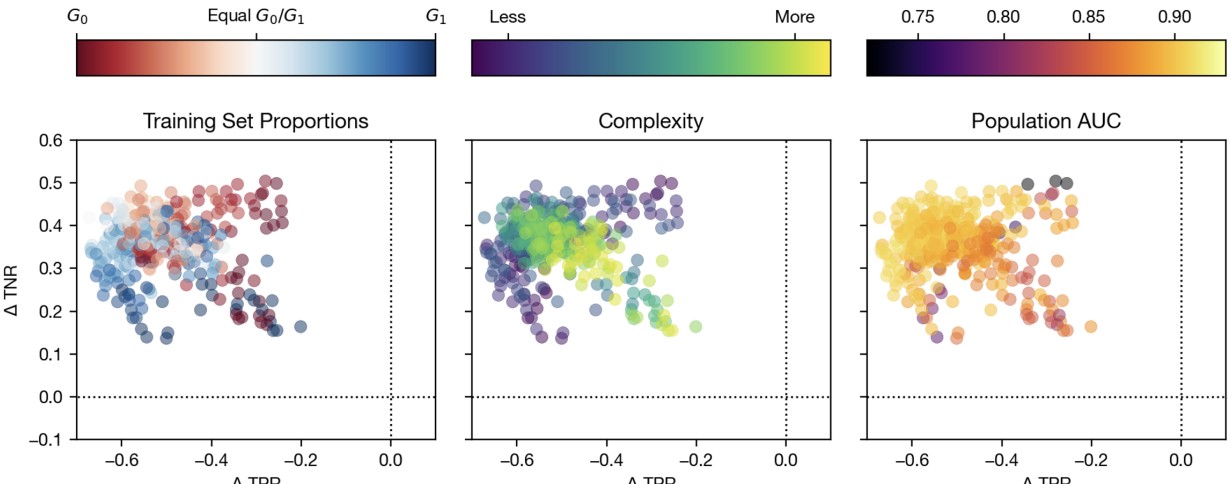

Figure 39: TPR and TNR unfairness at varying levels of group representativeness and complexity for the Community Crime data domain analyzed by race. Optimal fairness is shown by the intersection of the two dotted black lines. The leftmost panel is color-coded by group representation (deepest red indicating primarily $G_0$, deepest blue indicating primarily $G_1$, and white indicating balance. The middle panel is color-coded by model complexity, with lighter values indicating higher model complexity. The rightmost panel is color-coded by population AUC, with lighter values indicating higher performance.

