# OpenReview forum: "Disconnects between Dataset Representativeness and Group Algorithmic Fairness"
_TMLR — Withdrawn by Authors_

### Review · Reviewer_5iJW · 2025-02-18

**Summary Of Contributions:**

This paper contributes significantly to the understanding of algorithmic fairness and dataset representativeness by establishing a fundamental tradeoff between representativeness and group fairness, challenging the assumption that increased representativity inherently improves fairness. The authors theoretically and empirically validate this tradeoff in both simple and realistic settings. They model realistic dataset construction using a multi-armed bandit framework and a Bayesian approach, showing that practical sampling techniques complicate the representativeness-fairness relationship and that oversampling underperforming groups may not work in practice. Additionally, the paper identifies label difficulty as a key driver of unfairness and demonstrates that increasing model capacity can enhance fairness independently of representation. Finally, the authors propose a novel, representation-independent method to improve algorithmic fairness, offering a new approach to addressing group unfairness.

**Audience:**

Yes

**Claims And Evidence:**

Yes

**Requested Changes:**

See weakness.

**Strengths And Weaknesses:**

## Strengths

This paper makes several key contributions to the understanding of algorithmic fairness and dataset representativeness:
- **Tradeoff between representativeness and fairness**: The authors establish a fundamental tradeoff between dataset representativeness and group fairness, challenging the conventional wisdom that increasing representativeness inherently improves fairness. They theoretically and empirically demonstrate this tradeoff in simple and realistic settings.
- **Dataset Construction**: Using a multi-armed bandit framework and a novel Bayesian approach, the authors model the process of constructing representative datasets from multiple data sources. They show that realistic sampling techniques complicate the relationship between representativeness and fairness, revealing that oversampling underperforming groups—a theoretically sound solution—may not be effective in practice.
- **Label difficulty as a driver of unfairness**: The paper identifies that the difficulty of predicting labels for certain groups is a key driver of unfairness. It demonstrates that increasing model capacity can improve group fairness independently of dataset representation.
- **Representation-independent fairness improvement**: The authors propose a method to enhance algorithmic fairness that does not rely on adjusting dataset representativeness, offering a new direction for addressing group unfairness.

Overall, I find the topic of this paper interesting and commend the authors for their research on this subject. Additionally, I believe that the issue of dataset representativeness is crucial in the context of algorithmic fairness. The experiments are also very extensive, and the dataset construction section is impressive to me. I appreciate the authors for these.



## Weakness

- **Proof technique of Theorem 1**: As a reviewer focusing more on the theoretical part, I find that the proof technique for Theorem 1 appears insufficiently rigorous. Specifically, the authors claim that "the classifier with the highest accuracy on data (X,Y) will have the propriety that..." However, they do not provide a general solution for the optimal classifier. A more comprehensive analysis, including a derivation of the general form of the optimal classifier, would strengthen the theoretical foundation of the paper.
- **Simplistic problem setting**: The authors analyze the problem using a simple univariate setting to facilitate understanding. While this approach is reasonable for initial exploration, it raises questions about the generalizability of the results to more complex scenarios. For example, could the findings be extended to multivariate settings or other realistic cases? In the fairness literature, similar simplifications (e.g., linear regression settings [1]) have been employed to derive rigorous theoretical insights, which could serve as an example for extending this work.
- **Discussion of conventional fairness definitions**: In the experiments, the authors demonstrate group differences in AUC, TPR, and TNR. To enhance the paper's accessibility and relevance to the fairness community, it would be beneficial to provide a more detailed discussion on how the fairness definitions used in this work relate to widely recognized notions in the field. For instance, could the authors elaborate on the connections between their fairness criteria and established definitions such as Demographic Parity (DP), Equalized Odds (EO), and others presented in [2], as well as the principles of independence, separation, and sufficiency outlined in [3]?
- **Relevant theoretical results**: The paper could benefit from referencing additional theoretical results [4-5] that highlight the importance of base rates across different sensitive groups. These works are also somewhat relevant to the discussion.
- **Relevant algorithm**: As far as I know, there are some popular fair sampling approaches [6-7]. Maybe the author can consider conduct experiments on these two algorithms in the corresponding sections.

Minor issue:
- The authors use classifiers without fairness algorithms, such as constraints or regularization. Experimental results incorporating fairness-aware algorithms (such as pre-processing, in-processing, and post-processing methods [8]) could be an interesting addition and may further enhance the findings of this paper.
- At the top of Page 5, there is a typo in the notation: $n_1$



[1] Removing Spurious Features can Hurt Accuracy and Affect Groups Disproportionately. ACM FACCT 2021.

[2] A Survey on Bias and Fairness in Machine Learning. arXiv:1908.09635.

[3] Fairness and machine learning: Limitations and opportunities. MIT press, 2023.

[4] The Implicit Fairness Criterion of Unconstrained Learning. ICML 2019.

[5] Understanding Fairness Surrogate Functions in Algorithmic Fairness. TMLR 2024.

[6] Fairbatch: Batch selection for model fairness. ICLR 2021.

[7] Sample Selection for Fair and Robust Training. NeurIPS 2021.

[8] Fairness in Machine Learning: A Survey. arXiv:2010.04053

---

### Review · Reviewer_5PMt · 2025-03-08

**Summary Of Contributions:**

This paper considers the relationship between dataset representativeness and group algorithmic fairness, presenting that there exists a fundamental tradeoff between the two. The authors claim that improving representativeness does not necessarily lead to increased fairness by theoretical analysis, empirical validation, and sampling strategy evaluations. Additionally, the paper suggests that increasing model capacity can mitigate fairness issues independently of representativeness.

**Audience:**

Yes

**Claims And Evidence:**

Yes

**Requested Changes:**

see weaknesses

**Strengths And Weaknesses:**

Strengths:
1. The paper considers the widely held belief that improving dataset representativeness naturally leads to better group fairness.
2. The authors support their claims using a combination of theoretical proofs and empirical studies.
3. The paper highlights the challenges posed by real-world data collection methods.

Weaknesses:
1. The writing style is convoluted and lacks clarity, making it extremely difficult to understand the arguments, even after multiple readings. The paper attempts to discuss the relationship between dataset representativeness and group algorithmic fairness, but it is unclear why this topic is significant, what specific problem the authors aim to solve, and what their ultimate goal is. The theoretical proof section is especially confusing. For example, it lacks a clear logical structure, and its connection to the core claims of the paper is obscure. The lack of clear explanations and intuitive reasoning forces the reader to piece together the arguments, severely affecting the readability and impact of the work.
2. The paper lacks a coherent narrative structure, making it difficult to follow how different sections relate to each other. The theoretical and empirical sections feel disconnected, with little effort made to bridge the gap between them. The authors claim to explore fairness in real-world dataset construction, but the theoretical analysis is abstract and does not align well with the data collection scenarios studied in the experiments. The conclusions drawn in different sections are sometimes contradictory or weakly justified, leading to confusion about the key takeaways. The authors need to refine the organization of the paper to establish a more logical flow and ensure that theoretical insights are properly linked to experimental findings.
3. The motivation behind the study is poorly established. While the authors claim that dataset representativeness and fairness may be in conflict, they fail to clearly explain why this conflict is an important issue that requires attention. The paper lacks strong real-world examples or practical implications that justify why this problem deserves theoretical and empirical investigation. Furthermore, although the authors provide some high-level recommendations.For example, increasing model capacity to mitigate fairness issues,these conclusions are too vague and obvious. They do not offer any novel or actionable insights that could lead to meaningful improvements in fairness-aware algorithm design. If this claim were truly significant, the authors should have demonstrated how it could be leveraged to develop better fairness algorithms, making their conclusions more convincing and practical.
4. The theoretical component of the paper is underdeveloped and fails to capture the complexities of real-world fairness problems. The assumptions made in the theoretical framework are often unrealistic, limiting the practical relevance of the derived conclusions. The theoretical tradeoff between fairness and representativeness is only demonstrated in a toy example, which is insufficient to support the broad claims made in the paper. Additionally, the connection between the theoretical results and the main arguments of the paper is unclear, making it difficult to understand how the proofs contribute to the central discussion. The lack of clear intuition behind the theoretical findings further weakens the impact of this section.
5. The conclusions drawn in the paper are exaggerated and not well-supported by the evidence. The authors make strong claims about the existence of a fairness-representativeness tradeoff but fail to provide solid theoretical or empirical justification. The proposed method of increasing model capacity as a fairness-improving strategy is overly simplistic and ignores potential downsides, such as increased overfitting, reduced interpretability, or fairness degradation in certain settings. The paper does not offer practical guidance on how practitioners should balance fairness and representativeness in real-world applications, further diminishing its impact.
6. The empirical evaluation is weak and does not provide strong evidence for the arguments. The experiments rely on a small number of datasets and do not adequately test different fairness definitions or real-world fairness concerns, reducing the generalizability of the results. The study also lacks meaningful baseline comparisons against existing fairness-improving techniques such as reweighting, adversarial debiasing, or post-processing methods, making it unclear whether the claimed tradeoff is a novel insight or simply an artifact of the chosen methodology. Moreover, the experimental design does not effectively isolate the key variables to demonstrate the claimed fairness-representativeness tradeoff, and the results are not analyzed rigorously enough to establish strong causal relationships. If the claims about model capacity improving fairness were valid, they should have demonstrated how this insight could be used to modify existing fairness-aware learning algorithms. Instead, the experimental section merely reports fairness metrics without deriving meaningful algorithmic contributions.

---

### Review · Reviewer_eCPU · 2025-04-09

**Summary Of Contributions:**

This work investigates how group-representativeness in datasets affects group fairness. The authors show that the common practice of sampling the data distribution for improved representativeness does not always lead to fairer outcomes. The study:
- Theoretically demonstrates the trade-off between data representativeness and group fairness.
- Construct datasets using practical sampling methods to study the empirical impact of representativeness on group fairness.

**Audience:**

Yes

**Claims And Evidence:**

Yes

**Requested Changes:**

Intro
- The definition of fairness in the introduction should be rephrased so as not to misrepresent literature on fairness; group fairness is the definition adopted in this work

Sec 2
- In Definition 1, clarify that a is assumed to be binary.
- In Definition 2, should the group fairness definition be $U_{\neg a} - U_a$?

Sec 3
- In the Proof of Theorem 1, "a threshold classifier acting on both groups" seems to indicate using a shared threshold over both groups. However, the following equation describes a group-specific threshold. Please clarify this point to avoid any confusion.
- In the Proof of Theorem 2, the square root of the final equation should not cover $\sigma_0$
- The statement "This theorem indicates that, in order to limit the accuracy-disparity between groups to be no greater than δ, the number of samples collected from each group (n0, n1) cannot be too different." requires more justifications.

Sec 4
- This section and the introduction require more detailed justifications on why the collection is the proposed solution for representativeness. The introduction discusses how discarding data is not always desirable but would require further discussion on this point.
- The theory is based on a linear classifier assumption with $\mathcal{F}(x \mid g) = 1 \quad \text{if} \quad x \geq \frac{1}{n_g} \sum_{x_j \in \mathbf{X}_{\mid g}} x$. In this equation, shouldn't the sum depend upon $y$? The current classifier seems to be based on $x$ alone, which should not reflect an optimal classifier.

Sec 6
- What does it mean to have v to be uniform?
- In Fig3, for SRS why $G_1$ is not impacted despite a population with 100% $G_0$? How is the data evaluated in this scenario? Is it on the originally collected dataset?

Sec 7
- "Both fair arm-based and fair direct sampling are repeated 21 times in each train/test fold." - 21 is nonstandard; do you have any justification for this value?
- The fairness gap should be presented in this section.

Conclusion
- In conclusion, Chen 2018 does not recommend oversampling but collecting additional samples to improve fairness. The use of oversampling throughout may be confusing.

**Strengths And Weaknesses:**

*Strengths*
- Paper is clearly written
- Extensive experimentation
- Important study of common practice in fairness

*Weaknesses*
- The theoretical results are not directly verified. The authors should consider synthetic experimentations with control on the task complexity to verify the proposed theory.
- The paper assumes that the collected datasets are representative of the true underlying distribution. Such an assumption appears to be misguided, as some subgroups are often undercollected. In other words, the paper assumes that data are collected completely at random, and the proposed sampling proposes a collection at random (as defined in the missingness literature). Under such an assumption, the optimal data representation matches the underlying population, as shown in their experiments in which the optimal fairness gap is obtained when matching the original population. Recommendations to collect more data aim to correct for potential collection biases. A synthetic analysis in which the data collection is biased, and performances are computed on the underlying distribution would be of interest.
- The paper would benefit from clear recommendations on what should be done to fight biases.

---

### Note · Authors · 2025-05-08

**Comment:**

Per request, we are withdrawing this submission for revisions. We thank the editors and reviewers for their time and consideration of this manuscript.

**Withdrawal Confirmation:**

I have read and agree with the venue's withdrawal policy on behalf of myself and my co-authors.